# Exceptional loss in ozone in the Arctic winter/spring 2020

Jayanarayanan Kuttippurath[1]*, Wuhu Feng[2,3], Rolf Müller[4], Pankaj Kumar[1], Sarath Raj[1],
Gopalakrishna Pillai Gopikrishnan[1], Raina Roy[5]

[1]CORAL, Indian Institute of Technology Kharagpur, Kharagpur–721302, India.
[2] National Centre for Atmospheric Science, University of Leeds, Leeds, LS2 9PH, UK
[3] School of Earth and Environment, University of Leeds, Leeds, LS2 9JT, UK
[4]Forschungszentrum Jülich GmbH (IEK-7), 52425 Jülich, Germany
[5]Department of Physical Oceanography, Cochin University of Science and Technology, Kochi, India

*Correspondence to*: Jayanarayanan Kuttippurath (jayan@coral.iitkgp.ac.in)

**Abstract.** Severe vortex-wide ozone loss in the Arctic would expose both ecosystems and several millions of people to unhealthy ultra-violet radiation. Adding to these worries, and extreme events as the harbingers of climate change, exceptionally low ozone with column values below 220 DU occurred over the Arctic in March and April 2020. Sporadic occurrences of low ozone with less than 220 DU at different regions of vortex for almost three weeks were found for the first time in the observed history in the Arctic. Furthermore, a large ozone loss of about 2.0–3.4 ppmv triggered by an unprecedented chlorine activation (1.5–2.2 ppbv) matching the levels occurring in the Antarctic was also observed. The polar processing situation led to the first-ever appearance of loss saturation in the Arctic. Apart from these, there were also ozone-mini holes in December 2019 and January 2020 driven by atmospheric dynamics. The large loss in ozone in the colder Arctic winters is intriguing, and demands rigorous monitoring of the region.

## 1 Introduction

Apart from its significance of shielding the harmful ultra-violet (UV) radiation reaching the surface of earth, stratospheric ozone is a key component in regulating the climate (e.g. Riese, et al., 2012). Changes in stratospheric ozone are always a big concern for both public health and climate (WMO, 2018; Bais et al., 2019). Due to unbridled emissions of Ozone Depleting Substances (ODS) to the atmosphere since the 1930s stratospheric chlorine peaked in the polar stratosphere in the early 2000s (Newman et al., 2007; Engel et al., 2018; WMO, 2018). The first signatures of polar ozone loss appeared over Antarctica by the late 1970s (Chubachi et al., 1984; Farman et al., 1985), and it peaked to saturation levels in the late 1980s due to already high levels of stratospheric chlorine (Kuttippurath et al., 2018). Recent studies have demonstrated effectiveness of the Montreal Protocol and its amendments and adjustments in reducing halogen gases, with a corresponding positive trend in ozone in Antarctica (Salby et al., 2011; Kuttippurath et al., 2013; Solomon et al., 2016; Chipperfield et al., 2017) and in northern mid-

latitudes (Steinbrecht et al., 2014; Nair et al., 2015; Weber et al., 2018). However, a positive trend in the Arctic ozone is not reported yet possibly because of the large dynamically driven inter-annual variability of ozone there (Kivi et al., 2013; WMO, 2018).

Antarctic winters are very cold and the ozone hole is a common feature of these winters since the late 1970s. There were winters with very low stratospheric temperatures with a stronger vortex that showed relatively larger loss in ozone, such as the winters of 1996, 2000, 2003, 2006 and 2015 (Bodeker et al., 2005; Chipperfield et al., 2017). There were also winters with higher temperatures and smaller ozone losses as in the case of 1998, 2002, 2012 and 2019 (Müller et al., 2008; de Laat et al., 2010; Kuttippurath et al., 2015). Yet, the inter-annual variability of ozone loss in the Antarctic is very small in recent decades. On the other hand, colder winters with large losses of ozone (e.g. > 1.5 ppmv of loss) are rare in the Arctic (Rex et al., 2015, von der Gathen et al., 2021). The ozone loss derived from satellite and ozonesonde measurements show that most winters have ozone loss in the range of 0.5–1.5 ppmv and extremely cold winters showed large loss of about 1.5–2.0 ppmv (Manney et al., 2003; Kuttippurath et al., 2013; Livesey et al., 2015). Similarly, ground-based measurements show about 15–20% of loss in most Arctic winters, but the winters 1995, 1996, 2000, 2005 and 2011 were very cold with large loss of ozone, up to 25–30% (Goutail et al., 2005; Pommereau et al., 2018). However, these ozone loss values are still smaller than the 40–55% loss occurrence in the Antarctic (Kuttippurath et al., 2013; Pommereau et al., 2018).

The Arctic vortex is relatively short-lived (i.e. three to four months). The vortex normally strengthens by mid-December or early January and dissipates by mid-March. Major and minor warmings are common features of Arctic winters. The Arctic vortex in any winter would be frequently disturbed by planetary waves that emanate from the troposphere. In general, planetary wave numbers 1, 2 and 3 are mostly responsible for the momentum transfer to the stratosphere. This dynamical activity would increase the temperature in the lower stratosphere and trigger stratospheric warmings. The warmings can be minor or major, depending on the strength of wave activity, increasing the polar temperature and eventually disturbing the polar vortex. The vortex can be distorted, displaced, elongated and even split in two in accordance with the potency of momentum imparted by the waves. When the polar vortex is disturbed, the ozone loss will be smaller and the final warming can be as early as in late February or early March, as for many Arctic winters (e.g. Manney et al., 2003; Kuttippurath et al., 2012; Goutail et al., 2015). However, the vortex dissipates and chemical ozone loss terminates when a major warming occurs there. In an earlier study, Kuttippurarth et al. (2012) observed an increasing trend in major warmings, and ozone loss is found to be proportional to the timing of the major warmings, as early winter warmings stop polar stratospheric cloud (PSC) formation (i.e. stop the action of heterogeneous chemistry) because of the higher temperatures. This situation limits the activated chlorine available for ozone loss and results in smaller loss in warm Arctic winters. Since 1979, during the satellite era, there were two extreme winters with large loss of ozone in the Arctic; 2005 and 2011 (Coy et al., 1997; Feng et al., 2007; Horowitz et al., 2011). The occurrence of extreme events is a feature of climate change (e.g. IPCC, 2007). Therefore, the extremely cold winters with large loss in ozone could also be a harbinger of climate change. Previous studies have postulated that the cold winters will get even colder

with large loss in ozone (Sinnhuber et al., 2000; Rex et al., 2004; Chipperfield et al., 2005; Rider et al., 2013; von der Gathen et al., 2021). Analyses of the past colder Arctic winters indicate that it is likely that the colder winters may experience large loss in ozone, as in the case of 2005, 2016 and 2011. There are already studies on this winter discussing the ozone loss and meteorology (Manney et al., 2020; Wholtmann et al., 2020; Rao and Grafinkel, 2020; Weber et al., 2021; Innes et al., 2021; Wilka et al., 2021; Grooß and Müller, 2021; von der Gathen et al., 2021; Feng et al., 2021). However, in this study, we use different data sets, various ozone loss estimates methods, and several parameters together to study the polar processing and ozone loss in the Arctic winter 2020. This is particularly important as the winter was very cold in the stratosphere with the largest ozone loss in the observational record and experienced the total column ozone (TCO) values below 220 DU for several days in the vortex.

**2 Data and Methods**

We have used two satellite ozone profile datasets. The level 2 data from the

(i)        Microwave Limb Sounder (MLS) v4.2 ozone, ClO, $HNO_3$ and $N_2O$ measurements and

(ii)       Ozone Mapping and Profiler Suite (OMPS) v2.5 (ozone).

(iii)      We have also used the ozonesonde measurements from the Arctic stations at Alert (62.34° N, 82.49° W) and Eureka (79.99° N, 85.90° W).

Three satellite-based total column ozone (TCO) data are also employed (level 3) for our analyses

(iv)      Ozone Monitoring Instrument (OMI, DOAS v003),

(v)       OMPS (v2.1),

(vi)      Global Ozone Monitoring Experiment (GOME) 2 (GDP4.8),

(vii)     Modern-Era Retrospective analysis for Research and Applications (MERRA)-2 and

(viii)    Brewer spectrometers from Alert and Eureka.

These TCO measurements have an uncertainty of 2–5%. The ozone and other trace gas profiles are provided in pressure coordinates that are converted to isentropic coordinates using the temperature data from the same satellite, except for OMPS, for which the temperature data are taken from ERA5. We use the European Centre for Medium-Range Weather Forecasts (ECMWF) Reanalyses ERA5 potential vorticity (PV) on a 1°×1° grid to determine the vortex edge. The PV data are also converted to isentropic coordinates using the ERA5 temperature data. We computed the equivalent latitude at each isentropic level at 5 K intervals from 350 to 800 K, which is then used to compute the vortex edge using the Nash et al. (1996) criterion. We use measurements inside the polar vortex for the ozone loss analysis. The missing values in satellite measurements were filled with linear interpolation.

We have taken ozone, ClO, $HNO_3$ and $N_2O$ from the Aura MLS measurements. The ozone measurements at 240 GHz have a
vertical resolution of 2–3 km, vertical range of 261–0.02 hPa and an accuracy of 0.1–0.4 ppmv. The vertical range of $HNO_3$
measurements is 215–1.5 hPa, vertical resolution is 2–4 km, with an accuracy of 0.1–2.4 ppbv, depending on altitude. The
$N_2O$ measurements are available for the 68–0.46 hPa vertical range, and 68 hPa roughly equivalent to the 400 K isentropic
level. The data were extrapolated up to 350 K by performing exponential fitting to $N_2O$ vertical distribution at 400–600 K by
considering the exponential change of $N_2O$ with altitude. The accuracy of retrievals at 190 GHz is about 2–55 ppbv at this
altitude range and the vertical resolution is about 2.5–3 km. The vertical resolution of ClO measurements at 640 GHz is about
3–3.5 km over 147–1 hPa, and the accuracy of measurements is about 0.2–0.4 ppbv. The measurements also have latitude-
dependent bias of about 0.2–0.4 ppbv, depending on altitude (Livesey et al., 2013; Santee et al., 2008; Froidevaux et al., 2008).
The ozonesonde measurements have an uncertainty of 5–10% (Smit et al., 2007).

The OMPS consists of three sensors that measure scattered solar radiances in overlapping spectral ranges and scan the same
air masses within 10 min. The nadir measurements are used to retrieve ozone total column and vertical profiles (NP). The
Limb Profiler (LP) measures profiles with high vertical resolution ($\sim$ 2–3 km) and the LP retrievals are in good agreement
with other satellite measurements and the differences are mostly within 10% (Kramarova et al., 2018). The OMPS TCO shows
0.6–1.0% differences with Brewer and Dobson ground-based TCO measurements across the latitudes, and are also biased +2%
when the TCO is above 220 DU (Bais et al., 2014). GOME-2 was flown on MetOp-A satellite in 2006. The GOME-2 ozone
column has a positive bias in the northern high latitudes of about 0.5–3.5% (Layola et al., 2011). The OMI TCO measurements
have an accuracy of about 5% in the polar regions (Kroon et al., 2008; Kuttippurath et al., 2018). The Brewer spectrometers
operate in the UV region and their ozone observations have an accuracy of about 5%.

The ozone loss is estimated using two different methods and four different data sets to make sure the analyses are robust. The
first method used is the widely used profile descent method, wherein the $N_2O$ data are used for the calculations of air mass
descent in the polar vortex. The reference profile of $N_2O$ was taken from the month of December, and therefore, the loss
calculations are presented from December (May for Antarctic) onwards. The second method used for the calculation of ozone
is the passive tracer method, for which a passive odd-oxygen tracer is simulated using a CTM (Chemical Transport Model)
and is subtracted from the measured ozone to determine the ozone loss, as the changes in tracer are modulated only by the
dynamics (Feng et al., 2005). We have used the SLIMCAT model for the tracer calculations (Chipperfield, 2006) and
investigated the Arctic ozone loss under different meteorological conditions including Arctic winter/spring 2020 (e.g.,
Chipperfield et al., 2005; Bognar et al., 2021; Feng et al., 2021; Weber et al., 2021).

## 3. Results and discussion

### 3.1 The exceptional meteorology of the Arctic winter/spring 2020

Figure 1 shows the times series of stratospheric meteorology in the Arctic winter/spring 2020 compared to that long-lasting polar vortex years 1997 and 2011. Time series of the meteorological parameters for all Arctic winters since 1979 are also shown (grey coloured curves) for comparison. In general, the temperatures are between 210 and 195 K. In 2020, the temperatures were about 195 K in December, 190–195 K in January–March and 195–205 K in April. However, the minimum temperature in late winter 2020 is generally lower than 195 K, lasting about 115 days from December through early April. The temperatures are lower than those in the 2011 winter, and those in late March and April are the lowest on the observational record. The lower temperatures in late December through mid-March are key to PSCs, chlorine activation, the maintenance of high values of active chlorine and ozone loss. Low temperatures are thus a common phenomenon in winters with large loss of ozone (e.g. 1995, 2000, 2005 and 2011). Therefore, the higher temperatures in early winter and limited chlorine activation were the reasons for relatively smaller ozone loss in 1997; although it was a winter with a strong vortex up to the end of April (Coy et al., 1997; Feng et al., 2007; Kuttippurath et al., 2012). Since minor warmings (mWs) are very common in the Arctic winters, we also examined the occurrence of mW events by checking the temperature at 90° (North Pole) and 60° N at 10 hPa and zonal winds at 60° N at 10 hPa. The analyses show a small increase in temperature on 5 February 2020 (i.e. a minor warming) and a corresponding change in zonal winds.

The temperatures were consistently lower than the nitric acid trihydrate (NAT) equilibrium threshold of about 195 K and therefore, large areas of Polar Stratospheric Clouds (PSCs) are observed from December to mid-February. Even though PSCs may also be composed of liquid particles and not only NAT (e.g. Pitts et al. 2009; Spang et al., 2018), the NAT equilibrium threshold constitutes a good estimate for the occurrence of heterogeneous chemistry (e.g. Grooß and Müller, 2021; von der Gathen et al., 2021). The potential PSC area (APSC) was about 4 million $km^2$ in December 2020 at 460 K, but it doubled in January through mid-March. The APSC from mid-February to late March is also largest on the observational record (Figure 1). The low temperatures (i.e., lower than 188 K) also produced a very high amount of ice PSCs at the end of January and early February (up to 4 million $km^2$) when the lowest temperatures in 40 years were recorded in the Arctic. This is the largest ice PSC ever observed in terms of its area, volume and number of days of appearance (i.e. frequency) in the Arctic, and the area is twice that of the winter 2011 (also see Deland et al., 2020). The PSC area shrunk to half of its area in late January and February, as the lower stratospheric temperature increased during the period. This was the only occasion that the temperature increased and PSC areas limited to below 4 million $km^2$ in the winter 2020. Note that the PSC area and volume were largest in 2016, not in 2020 (Figure S1) (Kirner et al., 2015).

The potential vorticity (PV) at ~17 km (about 460 K potential temperature level) show that the polar vortex was very strong in the lower stratosphere in 2020. The PV values were consistently higher than the previous cold (i.e. 1995, 2000, 2005 and 2011) and long lasting (e.g. 1997 and 2011) winters in March and April. This indicates that the winter 2020 had the strongest

vortex in recent history, as demonstrated by the PV time series of different Arctic winters (Figure 1, second panel, left). However, the zonal winds were strongest in 1997 during the March-April period. The diagnosis with net heat flux and the eddy heat flux associated with planetary waves 1, 2 and 3 demonstrate that the momentum transported from the troposphere to stratosphere was very weak in 2020 (in the range of -20 to 30 Km s$^{-1}$), and the net heat flux values are zero or negative (e.g. -10 Km s$^{-1}$ in February) during most part of the winter. These results are also in agreement with the eddy heat flux computed for the waves 1–3, as they also show smaller wave momentum to the stratosphere. In short, the net heat flux and wave 1–3 heat flux show smaller values in January-April; indicating the reason for the less disturbed long-lasting vortex in 2020. According to Lawrence et al. (2020), apart from the weak tropospheric forcing, the formation of reflective configuration of stratospheric circulation was another factor that aided in the strengthening of the vortex in 2020.

The potential vorticity analyses show a strong and large vortex in early December. The vortex began to grow and occupied the entire polar region (defined by PV vortex edge) by early January, as shown in Figure 2. The lowest temperatures of the past 40 years were recorded by the end of January and the vortex was exceptionally strong and large (e.g. Wohltmann et al., 2020; Rao and Garfinkel, 2020). The mW distorted and elongated the vortex in early February, but the vortex was still strong and continued to be intact until the last week of April 2020. The extraordinary persistence of a strong and undisturbed Arctic vortex in March and April is evident in the PV maps. We also examined the Arctic winters since 1979 in terms of their dynamical activity, as shown in Figure S1b. The analyses show that, although the average vortex temperature and vortex area at 70 hPa was not very exceptional, the westerly winds (25 ms$^{-1}$) were strongest and dynamical activity was weakest (with heat flux 17 K ms$^{-1}$) in the past twenty years. This further suggests that the winter 2020 was unique and that wave forcing was very weak during the period.

### 3. 2 Strong air mass descent and associated ozone distribution

Figure 3 shows the distribution of ozone, ClO, N$_2$O, HNO$_3$ and the ozone loss estimated for the winter 2020 using satellite observations. We use the measurements from MLS on the Aura satellite (Livesey et al., 2015). The MLS data has been widely used for the study of polar ozone loss, as the instrument provides measurements of some key ozone-related chemistry trace gases such as ClO, N$_2$O and HNO$_3$ to delineate the features of chlorine activation, vortex descent and denitrification, respectively (Manney et al., 2020). The ozone distributions in the vortex show < 1.0 ppmv in December, slightly higher values of about 1.5 ppmv in February and smaller than 1.0 ppmv from mid-March to the end of April at 400 K. The measurements show exceptionally low values of ozone, about 0.5 ppmv or below, during the period mid-March through to the end of April at 350–450 K. The ozone values show < 2.5 ppmv from December to mid-January, < 2 ppmv January and February and < 1.0 ppmv in March-April at 350–450 K, and about 2–4 ppmv above 500 K; suggesting an unusual chemical depletion of ozone in December and late January. The ozone values are about 3–4 ppm above 550 K throughout the winter; implying little reduction in ozone there. The unusual feature here is the extremely small ozone mixing ratios of 1.0 ppmv in early December and March–April below 450 K (about 16 km). This reveals huge depletion of ozone in the lower stratosphere and therefore, we have

quantified the ozone loss for the winter. We estimate the descent rate from the tracer $N_2O$ inside the polar vortex, then assume
the averaged profile descent rate is identical to the dynamical ozone tracer so that the chemical ozone loss can be derived (e.g.,
Griffin et al., 2019). This is a widely used method for chemical ozone loss estimation (Rex et al., 2002; Jin et al., 2006).

For instance, the MLS measurements show that $N_2O$ values were 250 ppbv at 400 K, 150 ppbv at 500 K and 50 ppbv at 600
K in December. The $N_2O$ observations show strong air mass descent with values down to 100 ppbv at 400 K and about 25–50
ppbv above 500 K in early February. Again, $N_2O$ values exhibit below 50 ppbv in late March at 400 K. The $N_2O$ distributions
show below 50 ppbv at all altitudes from early February onwards; suggesting substantial dynamic descent in the stratosphere.
When a particular altitude is considered, e.g. the 450 K potential temperature level, the $N_2O$ values show 160 ppbv in early
December, 100 ppbv in early January, 50 ppbv in early February and less than 50 ppbv thereafter. On the other hand, the $N_2O$
distributions show 50 ppbv in early December and below that value afterwards at 500 K. The severe air mass descent in this
winter is further depicted in Figure S2, where monthly correlations between ozone and $N_2O$ are presented.

## 3.3 Ozone loss and mini-holes in December and January

There were vortex-wide PSC occurrences in the first week of December, about 2–4 million $km^2$ in area (APSC) and about 70
million $km^3$ in volume (VPSC) (see Rex et al., 2005 for the definitions). The APSC and VPSC dropped significantly afterwards
and then gradually increased again by mid-December to 10 and 120 million $km^3$, respectively. An unusual increase in activated
chlorine is observed during the first week of December in conjunction with the appearance of PSCs. The temperatures began
to decrease from 198 K in mid-December to 187 K by the end of January, as shown in Figure 1. The chlorine activation peaked
and showed record levels of ClO, about 1.5–2.0 ppbv at 400–600 K, during this period. The chemical ozone loss began in early
January with about 0.5 ppmv and increased to 1.5 ppmv by the end of January below 500 K. The loss above that altitude is
always lower than 0.5 ppmv, which shows that the ozone loss is restricted to the altitudes below 21 km (i.e. 550 K).

In general, the ozone loss starts in December in the middle stratosphere and then gradually progresses towards the lower
stratosphere by January. The loss would be below 0.5 ppmv in December and about 0.5–1.0 ppmv in January in the lower
stratosphere in cold Arctic winters. However, in the Arctic winter 2020, the ClO and ozone loss show unusually high values
of about 1.5–2.0 ppbv and 1.5–2.0 ppmv, respectively. Since ozone loss of this scale requires sunlight and high levels of ClO,
and one would not expect substantial amounts of sunlight in the early winter Arctic vortex, the appearance of huge amounts
of ClO during this period is surprising. The only possibility to have such high-levels of chlorine activation is the displacement
of vortex to sunlit latitudes. The analyses of vortex position in early December and late January (Figure 2) reveal that the
vortex was at 55°–60° N. Therefore, a strong polar vortex, very low temperatures, large volumes of PSCs and shift of vortex
to the sun light part of mid-latitudes caused the unprecedented chlorine activation and ozone loss in the first week of December
and late January. This is similar as that of the Arctic winter 2002/03 (e.g. Goutail et al., 2005; Kuttippurath et al., 2011).

In addition to the ozone loss inside the vortex, there is another interesting phenomenon in December and January. The analyses of TCO show that there were Arctic ozone mini-holes (e.g. Stenke and Grewe, 2003; Rieder et al., 2013) of about 300–700 km$^2$ size in the first week of December (1–6 December 2019) and on 26 January 2020 (Figure 4). The lowest TCO measured of the winter was also at the latter date. A detailed analysis with TCO, PV, temperature and ClO reveals that those ozone mini-holes were dynamically driven, as there was rapid air mass transport to the southern Arctic in early December and late January. These ozone mini-hole occurrences due to rapid changes in weather patterns and the total column ozone returns to the amount of normal levels of ozone in few days.

Ozone mini-holes are a dynamically driven sporadic decrease in TCO observed mostly in the mid-latitudes of both hemispheres due to rearrangement of the ozone column associated with tropospheric weather systems (Reed, 1950). The mini-holes are called so, as the TCO is less than 220 DU in those areas, and is one of the criteria defining the Antarctic ozone hole, although they differ in the nature of formation and spatial extent. These transient spatial and temporal events were identified first by Dobson and Harrison (1926) much before the identification of chemical ozone loss and were referred to as mini-holes by Newman et al. (1988) and McKenna et al. (1989). The plunge in TCO results when, the horizontally advected ozone poor tropospheric air mass interacts with the vertical air column motions in the anticyclonic ridging regions of the upper troposphere in the polar regions. As a consequence of this divergence, mixing or both may result in the appearance of mini-holes (e,g. Peters et al., 1995; James et al., 1997; Canziani et al., 2002). Since its identification, the criteria for the definition of mini-holes differed based on the thresholds of TCO amounts and spatial coverage in different geographical locations (Millán and Manney, 2017). In our study the threshold is taken to be 220 DU (see Bojkov and Balis, 2001). Many studies have also analysed the mini-hole formations in the northern hemisphere (e.g. James, 1998; Krzyścin, 2002; Stenke and Grewe, 2003; Feng, 2006). Here, we analyse the ozone mini-holes that appeared in the polar region of the winter 2020 and their dynamical origin.

We used the HYSPLIT trajectory model to find the air mass transport at three different altitudes (17, 18 and 19 km) in the lower stratosphere, where the mini-holes are found (Figure 4, right panels). The air mass exported from mid- and low latitudes has very low PV values, low temperature and high ClO. It suggests that the ozone transported from mid-latitudes triggered the ozone "holes" (ozone values < 220 DU). To further examine the low ozone values outside the vortex, we selected two ozonesonde measurements in the region (Alert: 62.34° N, 82.49° W and Eureka: 79.99° N, 85.90° W), which are shown in bottom panels of Fig. 4 for selected dates in December and January. These measurements show significant reduction in ozone (Coy et al., 1997; Feng et al., 2007; Horowitz et al., 2011) between 12 and 18 km; confirming the findings from the satellite total column measurements. Note that similar ozone mini-hole occurrences with comparable TCO, very low temperatures with huge VPSCs and high ClO in the mini-holes were also reported in some previous Arctic winters (e.g. Weber et al., 2002; Feng, 2006).

It should be mentioned that there was already large chemical loss of ozone inside the Arctic vortex in early December and late
January owing to the conventional polar ozone loss chemistry (as shown in Figure 3). However, the ozone mini-holes that
appeared outside the vortex were primarily caused by dynamics. We cross-checked TCO from OMI (Bias et al., 2014), OMPS
(Flynn et al., 2014), GOME-2 (Layola et al., 2011) and MERRA-2 (Gelaro et al., 2017), and found that the ozone mini-holes
were present in all these TCO datasets.

## 3.4 Prolonged chlorine activation and chemical ozone loss

When the Arctic winters are very cold, chlorine activation occurs in the Arctic lower stratosphere at 400–500 K in January and
February. In 2011, the chlorine activation was observed up to the end of February and was intermittent with a peak value of
about 1.6 ppbv, and was mostly at 400–500 K (e.g., Manney et al., 2011; Kuttippurath et al., 2012; Livesey et al., 2015; Griffin
et al., 2018). Conversely, in the Arctic winter 2020, there was continuous and sustained chlorine activation from December to
early April, except during the mW periods of mid-December and early February. The ClO values are also 0.5 ppbv larger than
those observed in the winter 2011. Feng et al. (2021) also stated that the chlorine activation in 2020 lasted longer than that in
2010/11. Therefore, strong chlorine activation was observed in March-April with ClO values of about 1.0–1.6 ppbv at 400–
550 K and the peak ClO value is about 2.1 ppbv.
The minor warming (Figure 1) caused a break in chlorine activation (Figure 3 for ClO) in early March. Nevertheless, the
temperature decreased shortly thereafter, which produced continued chlorine activation until early April at 400–550 K. The
ozone loss deepened in March and peaked by the end of March, and showed the maximum of about 1.5–3.4 ppmv at a broader
altitude range, up to 500 K. The ozone loss above that altitude (i.e. 550 K) was about 0.5–1 ppmv, which is still larger than
that that of any other Arctic winter. In fact, the loss of 1.0 ppmv is the peak loss observed in normal or moderately cold winters
of the Arctic (e.g., Kuttippurath et al., 2013); suggesting the severity of ozone loss even at the higher altitudes in this winter.
The maximum loss in 2020 was recorded at the end of March to the end of April, about 2.0–3.4 ppmv at 400–500 K and about
0.5–1.5 ppmv at 500–600 K. Furthermore, when compared to the early winter values, the late winter low $HNO_3$ values suggest
very severe denitrification, about 2–4 ppbv in the same period at 350–450 K (e.g. Manney et al., 2020). The $HNO_3$ values in
the lower stratosphere in March–April are about 60–80% lower than those of December–February at the same altitude levels
(Pommereau et al., 2018; Lindenmaier et al., 2012). The gravest denitrification was in December, with values of about 0–2
ppbv below 400 K and 4–6 ppbv at 400–450 K. Therefore, high chlorine activation and strong denitrification (as deduced from
the $HNO_3$ analyses shown in Figure 3) provided the basis for an unprecedented situation for large ozone loss of about 2–3.4
ppmv in the lower stratosphere in March–April.
Since the ozone loss in 2020 is exceptionally larger, we have employed another set of measurements to estimate ozone loss to
reconfirm that the derived results are robust. The loss estimated from OMPS measurements together with other analyses are
shown in Figure 5 (left). The maximum ozone loss profile extracted from the OMPS data shows very good agreement with
that from the MLS measurements for the Arctic winter 2020. The peak ozone loss values show about 2–2.8 ppmv in the lower

stratosphere below 550 K. Since the maximum ozone loss profiles are averaged for a few days, the loss values are slightly lower than those from MLS. The lower stratosphere shows similar ozone loss values, but the loss above 500 K shows slightly smaller values (0.1–0.5 ppmv) due to the low bias of OMPS measurements at these altitudes as compared to the MLS measurements (Kramarova et al., 2018). The comparison with OMPS confirms that the method adopted for ozone loss is robust. Our estimates are in good agreement with those of Manney et al. (2020), Weber et al. (2021) and Wohltmann et al. (2020), who also derive a loss of about 2.1–2.8 ppmv below 450 K from the MLS measurements.

## 3. 5 The Arctic ozone loss in the context of other Arctic winters

Arctic winters are normally warmer that those in the Antarctic and occurrences of PSCs are sparse and infrequent. Therefore, high chlorine activation and significant ozone loss are limited to winters with very low temperatures in December–February (Tilmes et al., 2014; Goutail et al., 2005; WMO, 2018; Newman et al., 2008; Kuttippurath et al., 2012). The ozone loss observed in warm winters (e.g. 2006 and 2009) is about 0.5–0.7 ppmv, moderately cold winters (e.g. 2008 and 2010) is about 1.0–1.2 ppmv and very cold winters (e.g. 2005) is 1.4–1.6 ppmv (e.g. WMO, 2018). However, the ozone loss in the winter 2011 was about 1.0 ppmv (or 30–40 DU) larger than that of other Arctic winters (about 2.1–2.3 ppmv or 100–100 DU). This ozone loss was similar to the loss found in warmer, more perturbed Antarctic winters (e.g. 1988 and 2002) (Manney et al., 2011; Kuttippurath et al., 2012; Feng et al., 2015; Pommereau et al., 2018). We applied the same loss estimation method to the measurements for the Arctic winter 2011 to compare with that of the Arctic winter 2020. This would also test the veracity of the loss estimation procedure and the results are shown in Fig. 5.

The peak ozone loss in the Arctic winter 2011 is about 2.1 ppmv, which is in very good agreement with all other available analyses for that winter (WMO, 2014, 2018; Griffin et al., 2018; Livesey et al., 2015). However, the ozone loss in the Arctic winter 2020 is about 0.7 ppmv higher than that in 2011, about 2.8 ppmv. The difference in ozone loss between the winters is negligible above 480 K. Therefore, it is evident that the ozone loss in the Arctic winter 2020 is the largest on the record and is significantly higher than that of any previous Arctic winter (Grooß and Müller, 2021).

Furthermore, we applied another loss estimation method to test robustness of the extreme ozone loss values; the passive method that uses a passive tracer (i.e. no chemistry) simulation. We have used the well-known and widely used TOMCAT/SLIMCAT model simulations for the tracer calculations (Chipperfield et al., 1999; Dhomse et al., 2019). The ozone loss computed with the passive method shows the peak value of about 2.3–2.5 ppmv at about 450 K in the Arctic winter 2020 (Figure 5, second panel from the left). This ozone loss is slightly higher than that of the Arctic winter 2011, about 0.2 ppmv. It is also observed that the ozone loss in 2020 is higher than that of 2011 below 475 K, but the loss estimated in the 2011 winter exceeds about 0.3–0.5 ppmv above 475 K up to 700 K (e.g. Manney et al., 2020; Wohltmann et al., 2020). However, these ozone loss estimates are lower than those estimated with the descent method, by about 0.5–0.7 ppmv, depending on altitude. The analysis with ozone and $N_2O$ from the model indicates that modelled ozone is higher than (by about 1–1.5 ppmv) the measurements at these altitudes, which could be due to the slower dynamical descent in the model.

It is clear that the ozone loss in 2020 is the largest among Arctic winters so far. Therefore, we also examined the evolution of
chlorine activation in terms of the amount of ClO in each Arctic winter, as the total chlorine is decreasing in the stratosphere
due to the effect of the Montreal Protocol (e.g. Strahan et al., 2017; WMO, 2018; Dhomse et al., 2019) and we expect a
corresponding response in ozone loss in the polar winters. Stratospheric halogen levels (EESC) in the Arctic in 2020 are more
than 10% below the maximum levels in 2000 (Grooß und Müller, 2021). Figure 6 shows the MLS ClO observations, the
December-February and December-March potential PSC areas, and EESC in each winter since 2005. The analyses show that
the chlorine activation was very severe and continuous for about four months in 2020. However, the highest ClO and largest
APSC values were observed in winter of 2016. Many cold winters showed ClO values around 1.8–2.0 ppbv as found in 2020,
but the sustained chlorine activation that was observed in 2020 was unique. Although the high ClO values in March were also
observed in 2011, the chlorine activation was not as severe as in 2020 in early winter (December–January). The record-
breaking spatial extent of ice PSCs in the winter 2020 might have also contributed to the exceptional chlorine levels. On the
other hand, the unprecedented chlorine activation observed in 2016 was more episodic, such as in mid-December, mid-January
to early February and late February. Therefore, the continuous and severe chlorine activation from December through March
was the key for the record-breaking ozone loss in 2020. Figure 6(b) and (c) further illustrate that the peak ClO profiles or the
time series of average ClO for the entire winter will not reveal the depth of chlorine activation. We also looked at the changes
in EESC during the period (2005-2020) and there has been a continuous decline in EESC during the period (Fig. 6, top panel).
The predicted rate of change of EESC during the period is about 246.16 ppt per year (e.g. WMO, 2018); suggesting a reduction
in stratospheric halogen loading in 2020 compared to the peak loading by about 10% (e.g. Grooß and Müller, 2021).
**3.6 The Arctic ozone loss and the Antarctic ozone loss**
The peak ozone loss in the Antarctic happens at around 500 K and the loss is severe from 400 to 600 K for five months
continuously from August to November (Tilmes et al., 2006; Huck et al., 2005; Sonkaew et al., 2013; Kuttippurath et al.,
2015). In contrast, the cold Arctic winters are normally shorter and maximum ozone loss occurs at around 425–475 K for a
period of about two months, from mid-January to mid-March (e.g. Kuttippurath et al., 2010; Manney et al., 2004). The ozone
loss in the Arctic is limited to the altitudes below 500 K. The ozone loss in the Arctic winter 2020 was very high, and therefore,
we compare the Arctic ozone loss in 2020 with that in the Antarctic winters 2015 and 2019. The Antarctic winter 2015 was
one of the coldest and 2019 was one of the warmest, and therefore, the assessment would give an upper and lower bound of
ozone loss estimate for the Arctic winter 2020.
The peak ozone loss estimated using the vortex descent method is about 2.8 ppmv at 480 K in the Antarctic winter 2015 and
about 2.3 ppmv at 490 K in 2019 (Figure 5). The ozone loss in the Antarctic winter 2015 shows consistently higher values
(about 0.1–0.5ppmv) than that of 2019 up to 550 K, and the loss is similar above that altitude in both winters. The ozone loss
is about 1.0 ppmv at 370 K, 2.6 ppmv at 460 K, 1.5 ppmv at 550 K, 0.5 ppmv at 650 K and it terminates at 700 K in the

Antarctic winter 2015. In the Arctic winter 2020, the ozone loss shows about 0.3 ppmv at 370 K, 2.0 ppmv at 430 K and 480 K, 1.5 ppmv at 550 K and loss terminates above that altitude. The peak ozone loss is about 2.3 ppmv at 460–470 K. On the other hand, the loss in the Antarctic winters above 470 K is very large and reaching up to 700 K. The peak ozone loss in the Arctic winter 2020 is about 2.8 (2.3) ppmv and is at 460–470 K. This is also the main difference between the Arctic and Antarctic ozone loss, as the broader and larger ozone loss occurs above the 470 K in the Antarctic. The difference is almost 1.0 ppmv above the peak ozone loss altitude. Therefore, the ozone loss in the Arctic winter 2020 is either equal or larger than that of the Antarctic winter 2019 below 470 K, but the loss is smaller than that of the Antarctic winters above 525 K.

We have also applied the passive method to further examine the estimated loss in the Arctic and Antarctic winters (Figure 5, second panel from the left). The ozone loss estimated with the passive method exhibits smaller values in the lower stratosphere in comparison with that derived from the descent method. The loss is about 0.2 ppmv at 350 K, 1.6 ppmv at 400 K and 2.3 ppmv at 450 K in the Arctic winter 2020. The peak loss is recorded at 450–460 K and the loss decreases with altitude, about 1.5 ppmv at 500 K and 0.1 ppmv at 530 K. In the Antarctic winter 2019, the ozone loss shows similar values as that of the Arctic winter 2020 at 370–420 K, but slightly smaller than that of the Arctic winter at 420–470 K. The maximum ozone loss in Antarctic winter 2019 is estimated at 470 K, about 2.3 ppmv, and about 0.5–1.5 ppmv above that altitude, which is higher than that of the Arctic winter 2020. Furthermore, the Arctic ozone loss halts at about 550 K, whereas the Antarctic ozone loss at this altitude is as high as 1.5 ppmv.  In the Antarctic winter 2015, the ozone loss is about 1.0 ppmv at 370 K, 2.0 ppmv at 400 K and the peak loss of about 2.8 ppmv at 475 K. The loss gradually decreases with altitude, such as 2.1 ppmv at 500 K, 1.5 ppmv at 550 K, 1.0 ppmv at 600 K and 0.5 ppmv at 650 K. The diagnosed ozone loss in the Antarctic winter 2015 is thus, higher than that of the Antarctic winter 2019 and the Arctic winter 2020, by about 0.5–1.5 ppmv, depending on the altitude. The assessment further gives strong evidence that the peak ozone loss in the Arctic winter 2020 is similar to that of the warm winters of Antarctic (e.g. 2019). The loss estimation method can have uncertainty in the range of 3–5%, depending on the winter months. For instance, the monthly mean ozone loss and its standard deviation for each winter month of 2020 are shown in Figure S3. A complete error analyses of the passive method to estimate ozone loss is already presented in Kuttippurath et al. (2010).

**3.7 The first appearance of ozone loss saturation in the Arctic**

Ozone loss saturation (i.e. $O_3$ values less than 0.1 ppmv) is a common feature of Antarctic winters since 1987 (Jin et al., 1996; Solomon et al., 2005; Kuttippurath et al., 2018). However, as compared to the Antarctic, the Arctic winters are relatively short (Decembe–March), stratospheric temperatures are about 10 K higher, occurrence of PSCs are infrequent, denitrification is modest and thus, ozone loss is generally more moderate. Therefore, the Arctic never encountered the ozone loss saturation (i.e. the near complete (about 90–95%) loss of ozone at some altitudes in the lower stratosphere between 400 and 550 K) there before. Apart from these, the vortex-averaged ozone loss normally happens only up to 25–30% in the Arctic winters as analysed from ground-based spectrometer observations and henceforth, a loss saturation was unexpected for the Arctic conditions.

Figure 5 (right) shows the ozone profile measurements by ozonesondes at two Arctic stations, Alert (82.50° N, 62.33° W) and
Eureka (80.05 N, 86.42 W), on selected days. The ozone profiles measured at selected Antarctic stations are also shown for
comparisons. In general, the ozone loss saturation in Antarctica occurs at the altitude between 400 and 500 K (e.g. Davis:
68.6°S, 78.0°E and Marambio: 64° S, 56° W), and the altitude range would go up to 550 K for the stations that are always
inside the vortex, as shown for Syowa. Note that the ozone loss saturation is taken as 0.2 ppmv and ozone detection limit of
sondes is 10 ppbv (Kuttippurath et al., 2018; Solomon et al., 2005; Vömel and Diaz, 2010). The ozone loss observed at Davis
and Marambio is always smaller than that at Neumayer, South Pole and Syowa. Therefore, ozone loss saturation is also
different at different stations in the Antarctic. Here, the ozonesonde measurements at Alert (on 08 April 2020) show loss
saturation at the altitudes 420–475 K (e.g. Wilka et al., 2021). The measurements at Eureka (on 10 April 2020) show loss
saturation with about 99% ozone loss at altitudes between 420 and 460 K (see also Bognar et al., 2021). The time series of
ozone measurements, as analysed from the available measurements, show that the ozone loss saturation occurred at these
stations in early April (Figure S4). The vertical shading in Figure 5 for 0.2 ppmv shows the ozone loss saturation criterion with
respect to the ozone volume mixing ratios and the ozonesonde measurements have an uncertainty of 5–10% (Smit et al., 2007).
Yet, the ozone measurements at Alert and Eureka are in the saturation limit and thus, provide first evidence for the occurrence
of ozone loss saturation in the Arctic. The loss saturation suggests that the Arctic polar stratospheric has entered a new era of
change. Our analyses are consistent with the analyses of Wohltmann et al. (2020), who report about 90–93% loss of ozone in
the 450–475 K range in 2020 and with those of Grooß and Müller (2021) who find a lowest simulated ozone mixing ratio of
about 40 ppbv in 2020.
**3.8 Days with ozone values below a threshold of 220 DU**
Since the Antarctic ozone hole is defined with respect to TCO measurements (i.e. below 220 DU), we analysed TCO
measurements for the Arctic in 2020, which are shown in Figure 7. It shows the lowest TCO measurements made in the Arctic
polar region in the winter of 2020 by three different satellite instruments, OMI, OMPS and GOME. As shown (Fig. 7), the
OMI measurements show TCO below 300 DU for almost all winter months inside the vortex, as defined by Nash et al. (1996).
The measurements show around 230 DU in early December, about 260 DU in January, about 218–260 DU in February, around
220 DU in March and around 240 DU in April. There are ozone values lower than or equal to 220 DU in early (01–05)
December, late (25–26) January, some days (05, 12 and 17–22) in March and few days in early (06–07) April. The occurrences
of these low ozone values in December and January are associated with ozone mini-holes triggered by dynamics. However,
the appearances of extremely low TOC, below 220 DU, values in March and April are driven by chemistry and this is our topic
of discussion. The very low ozone measured by OMI corresponding to the dates are also shown in the ozone maps in the top
panel and the exact dates of extremely low ozone occurrences based on OMPS and MERRA-2 data are given in Table S1. The
OMPS total column agrees well with that of the OMI measurements throughout the period, where the differences are mostly
2–3 DU and are within the uncertainty of both instruments (i.e. about 5–10%). The OMPS measurements have captured all
features of OMI measurements throughout the winter. The GOME measurements are very close to the OMI and OMPS

measurements too, but are slightly higher in January and February due to the limited coverage of northern polar region by GOME in winter months. As the winter progresses, the GOME coverage improves and therefore, the March and April measurements are in excellent agreement with other satellite observations. The TCO measurements at Alert also manifest the low ozone values of about 200 DU in two days of April; corroborating the satellite observations (Figure 7).

We also estimated the partial column ozone loss from the ozone profiles of OMPS and MLS satellites (Figure 7, bottom panel). The ozone loss is calculated with respect to the passive method (Feng et al., 2005). The Arctic winters usually show TCO loss of about 70–80 DU in cold winters, about 45–50 DU in warm winters, and about 90–110 DU in exceptionally cold winters such as in 2005 and 2011 (Goutail et al., 2005; Kuttippurath et al., 2012b; Rex et al., 2005; Manney et al., 2003). The largest column ozone loss deduced hitherto was in the Arctic winter 2011, and was about 110 DU as assessed from all available studies (Griffin et al., 2018; Kuttippurath et al., 2012; Manney et al., 2011). On the other hand, the Antarctic ozone column loss is about twice that of the Arctic, about 150–160 DU, but slightly lower about 100–120 DU in warm winters (1988 and 2002) and in early years (e.g. 1979–1985) of ozone loss there (Huck et al., 2005; Tilmes et al., 2006; Kuttippurath et al., 2015). The analyses suggest that even the partial column ozone loss in the Arctic winter 2020 is about 115 DU at 350–550 K, which is higher than that of the Arctic winter 2011 and similar to that of the loss found in the Antarctic winters 1979–1985, 2002 and 2019.

Since the ozone loss in the Arctic winter 2020 is up to the levels of that found in some Antarctic winters, we examined the occurrence of extremely low TCO values using data from OMPS and MERRA-2; the results are presented in Figure 8 for selected days. The first appearance of ozone holes in Antarctic winters is also shown for comparison. There are clear and identifiable regions of extremely low TOC (regions below 220 DU) in March and April 2020, which were hundreds of kilometres wide (see also Dameris et al., 2021). The ozone maps show that the low ozone regions in March and April 2020 were larger than those measured in the Antarctic in October 1979 and 1980. Therefore, ozone loss in the Arctic winter 2020 is roughly comparable to the Antarctic ozone loss in 1980. The appearance of a threshold in TCO below 220 DU for several weeks demonstrates that Arctic winters may enter a new era of ozone depletion events (e.g. von der Gathen et al., 2021). However, extremely low TOC values neither appeared in all parts of the vortex nor are present continuously for months as they occur over the Antarctic; further, very strong chemical ozone loss occurs very regularly in the Antarctic, whereas strong Arctic ozone loss occurs only in very cold years (Bodeker et al., 2005; Tilmes et al., 2006; Feng et al., 2007; Müller et al., 2008; von der Gathen et al., 2021).

**4. Conclusions**

The Antarctic ozone hole has been present for the past forty years, and the impact of ozone hole on public health is mostly restricted to the southern high and mid-latitudes. The ozone hole has also influenced the climate of southern hemisphere by changing the winds, temperature and precipitation in different regions. On the other hand, the biggest concern about the polar ozone loss in the stratosphere has always been strong Arctic ozone loss, because such an ozone reduction can occur anywhere beyond 45° N in the densely populated northern mid and high latitudes. The changes in associated UV radiation incidence

would also affect the flora and fauna of the region. If such a situation arose, it would trigger ecosystem damage and impose a serious threat to public health (e.g. Newman et al., 2009). An account of the record-breaking increase in UV radiation in the 2019/20 Arctic winter is presented by Bernhard et al. (2020). Nevertheless, it is believed that extreme reductions in column ozone over the Arctic would be unlikely due to relatively higher temperature and a shorter wintertime ozone loss period there. Furthermore, Arctic winters are always prone to several minor and frequent major warmings (almost a major warming per winter), which would restrict the lifetime of the polar vortex, PSC occurrence and chlorine activation to limit the extent and severity of ozone loss. However, the Arctic winter 2020 was exceptional as it was characterised by a strong vortex from December through the end of April, large and widespread PSC occurrence, and unprecedented and prolonged chlorine activation with peak ClO values of about 2.0 ppbv. The high chlorine activation in early December and early January produced larger loss in ozone (e.g. 1–1.5 ppmv below 430 K in early January) in the Arctic that has never occurred before, consistent with the results of the studies of Weber et al. (2021) and Innes et al. (2020). The continued high chlorine activation from January to mid-April caused a record-breaking ozone loss of about 2.5–3.4 ppmv at 400–600 K, and triggered the first-ever observation of extremely low ozone columns in the Arctic in March and April 2020. The unprecedented chlorine activation (e.g. January through March, above 0.7 ppbv) and severe denitrification (60–80%) also set up the atmosphere to have the first ever occurrence of ozone loss saturation in the Arctic. Another interesting aspect of this winter was the dynamically driven but chemically modified ozone mini-holes in December and January. These mini-holes were larger than the Antarctic ozone holes of 1979 and early 1980s. The analyses presented use multiple data sets, different ozone loss estimation methods, and several parameters to make a robust statistics and a balanced assessment of the polar ozone depletion in the Arctic winter/spring 2020.

**Acknowledgements**

We thank Head CORAL, and the Director of Indian Institute of Technology Kharagpur (IIT KGP), Ministry of Human Resource Development (MHRD), and Naval Research Board (OEP) of Defense Research and Development Organisation for facilitating the study. PK acknowledges the support from MHRD and IIT KGP. GSG, SR and JK acknowledges the funding from DRDO OEP. We thank the data managers and the scientists who worked hard for making available the MLS, OMPS, OMI, MERRA, ER5, ozonesonde, GOME, and all other data for this study.  We also thank the HYSPLIT model developers for the trajectory analyses. The authors thank Paul Newman, Larry Flynn, Lucien Froidevaux, Jonathan Davies, Peter von der Gathen and Martyn Chipperfield for their help and support in making this article happen. The authors thank Martyn Chipperfield for his suggestions an comments on the manuscript. The SLIMCAT forced by ERA5 simulation was performed on the University of Leeds ARC4 HPC system.

**Code and Data availability**

The MLS data are available on https://disc.gsfc.nasa.gov/. The MODIS datasets were acquired from the Level-1 and Atmosphere Archive & Distribution System (LAADS) Distributed Active Archive Center (DAAC), located in the Goddard

Space Flight Center in Greenbelt, Maryland (https://ladsweb.nascom.nasa.gov/). The ozonesonde data are available from the
World Ozone and Ultraviolet Radiation Data Centre (WOUDC, https://woudc.org/). The OMPS ozone data are available on
https://earthdata.nasa.gov/earth-observation-data/near-real-time/download-nrt-data/omps-nrt. The meteorological analyses:
temperature, winds, heat flux, PSC and wave heat flux data are taken from https://ozonewatch.gsfc.nasa.gov/. The OMI data
are available on https://disc.gsfc.nasa.gov/datasets/. The GOME data are downloaded from the
https://atmosphere.copernicus.eu/data. The codes used for data analysis can be provided on request.

**Author Contributions**
JK conceived the idea and wrote the original manuscript. The manuscript was subsequently revised with inputs from RM and
WF. JK, PK, SR, RR and GK analysed the data and produced the figures. WF designed the model runs and carried out the
model simulations. All authors participated in the discussions and made suggestions, which were considered for the final draft.

**Additional information**
Supplementary Information accompanies the paper on this journal website.

**Competing Interests**
J. K. and R.M. are editors of ACP; otherwise, the authors declare no competing and conflict of interests

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

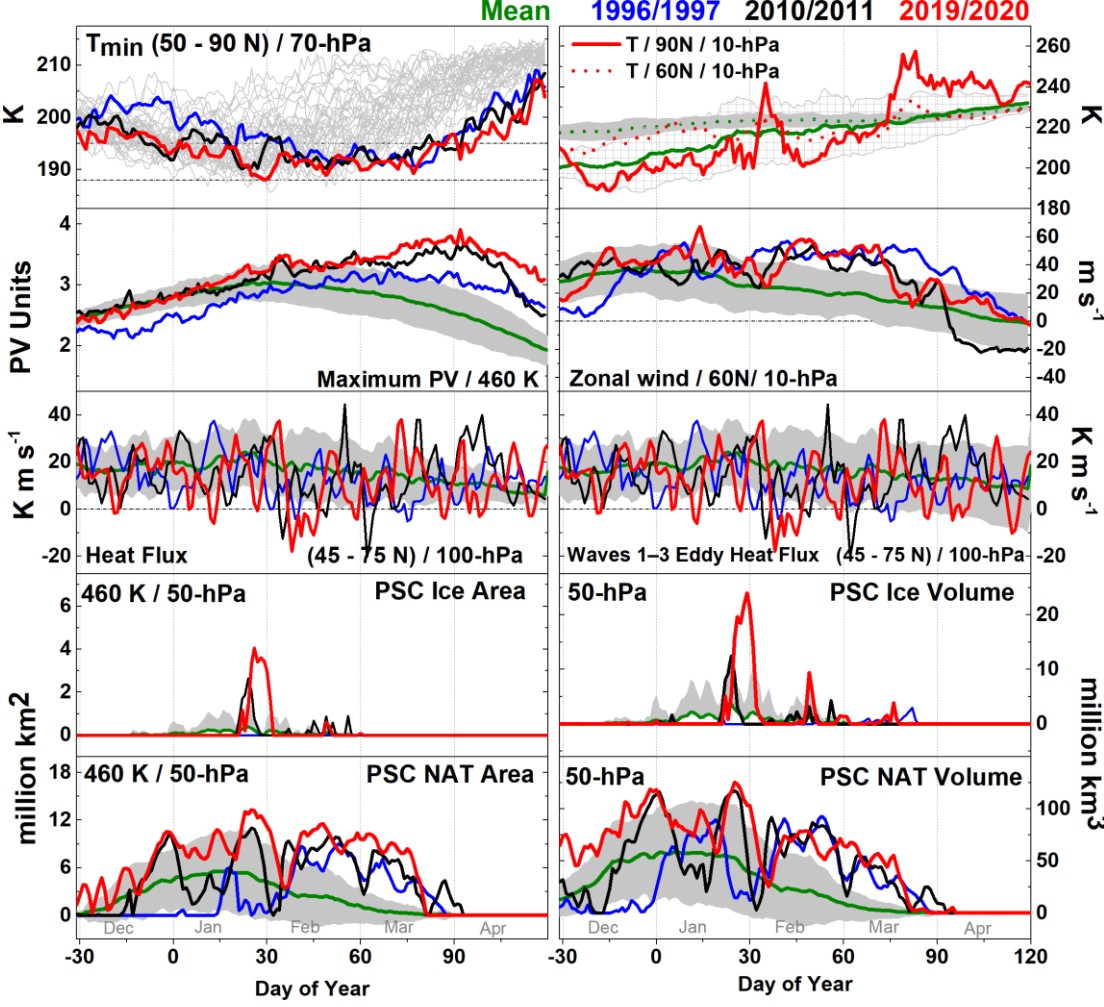

**Figure 1: Meteorology of the Arctic winter/spring 2020**. The temperature, zonal winds, potential vorticity (PV), heat flux, wave eddy heat flux, and area and volume of polar stratospheric clouds (PSCs) for the Arctic winter 2020 as compared to previous Arctic winters. The shaded area shows the standard deviation from the mean.



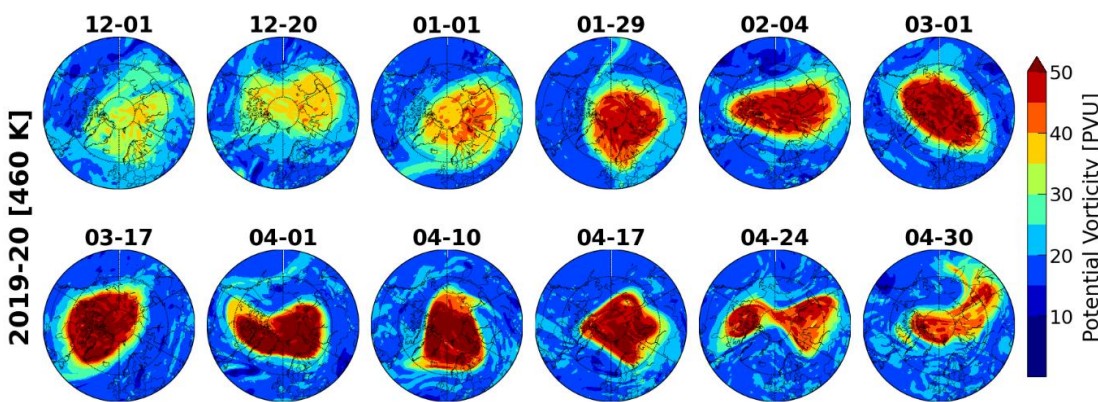


**Figure 2: Polar vortex evolution in the Arctic winter/spring 2020.** The evolution of polar vortex in the Arctic winter 2020. The vortex situation in the lower stratospheric altitude of about 460 K (~17 km) is illustrated. The vortex edge is calculated with respect to the Nash et al. (1996) criterion at each altitude.
























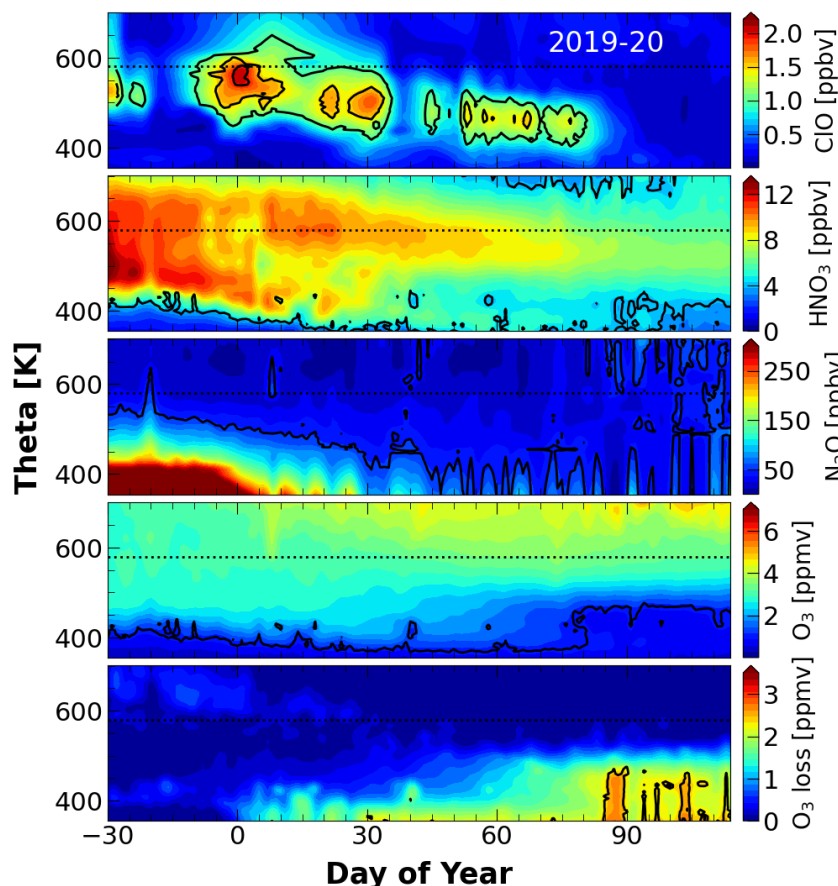


**Figure 3: Ozone loss in the Arctic polar vortex in 2020.** The distribution of ClO, HNO$_3$, N$_2$O and ozone (top to bottom) as measured by the Microwave Limb Sounder (MLS) for the Arctic winter 2020. The bottom panel shows the ozone loss estimated using the MLS ozone by applying the tracer descent method (see Methods and supplementary file). The vortex edge is computed in accordance with Nash et al. (1996) criterion. The vortex-sampled data are then averaged over each day and are shown.







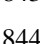

**Figure 4: The Arctic ozone mini-holes in December 2019 and January 2020**. The total ozone observations by Ozone Monitoring Instrument (OMI) on 03 December 2019 and 26 January 2020. The potential vorticity (PV) maps for the corresponding dates are shown on the right. The air mass trajectories computed using the HYSPLIT model at 17, 18 and 19 km are also illustrated in the PV maps. The ozonesonde measurements in December and January at Alert (62.34° N, 82.49° W) and Eureka (79.99° N, 85.90° W) are illustrated in the bottom panel and are also shown in the maps as red and magenta stars, respectively.

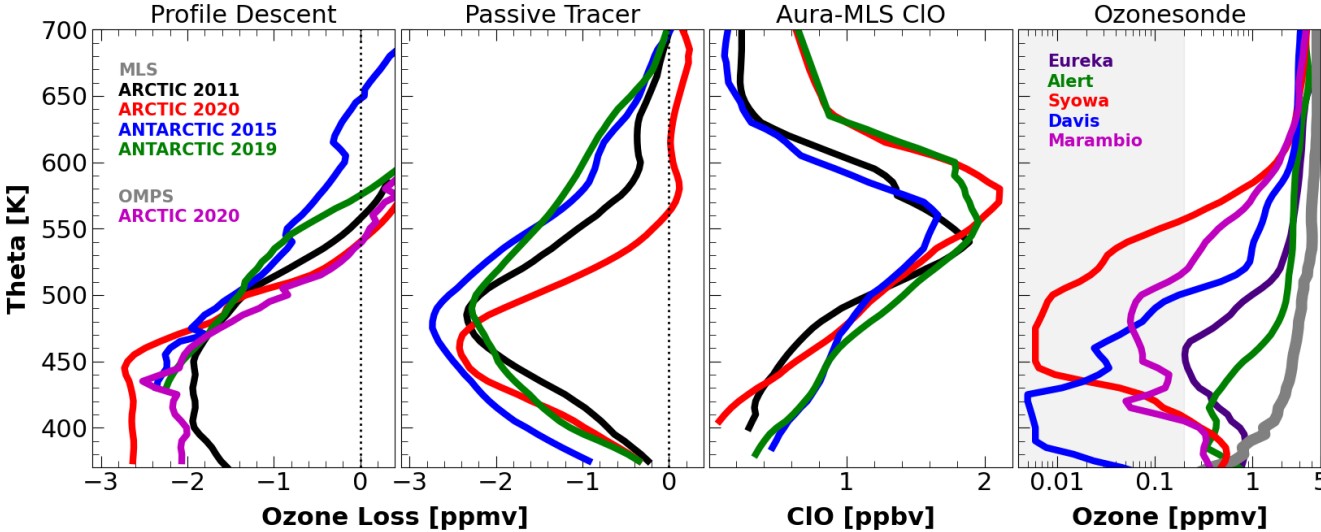

**Figure 5: The Arctic and Antarctic Ozone Loss Saturation and Chlorine activation. Left.** The ozone loss estimated using the Microwave Limb Sounder (MLS) measurements by applying the vortex descent method for the Arctic winter 2019–2020 compared to the Arctic winter 2011, and the Antarctic winters 2015 and 2019. The ozone loss estimated with Ozone Mapping and Profiler Suite (OMPS) measurements is also shown. **Second from the left**: The ozone loss estimated using the passive tracer method for the Arctic winter 2020, and the Antarctic winters 2015 and 2019. **Second from the right**: The activated profiles ClO measured by MLS for the Arctic winters 2011 and 2020, and the Antarctic winters 2015 and 2019. The profiles are selected for the days with peak ClO values and are averaged for three days. **Right:** Ozonesonde measurements from selected Antarctic and Arctic stations. The Antarctic ozonesonde measurements (Davis, Marambio and Syowa) from past winters and the Arctic measurements (Alert and Eureka) from the Arctic winter 2020. The grey colour represents an ozone profile without ozone depletion in Arctic and Antarctic. The grey-shaded region represents the ozone loss saturation threshold. The dates ozonesonde measurements are taken for 08 April 2020 (Alert) and 10 April 2020 (Eureka).

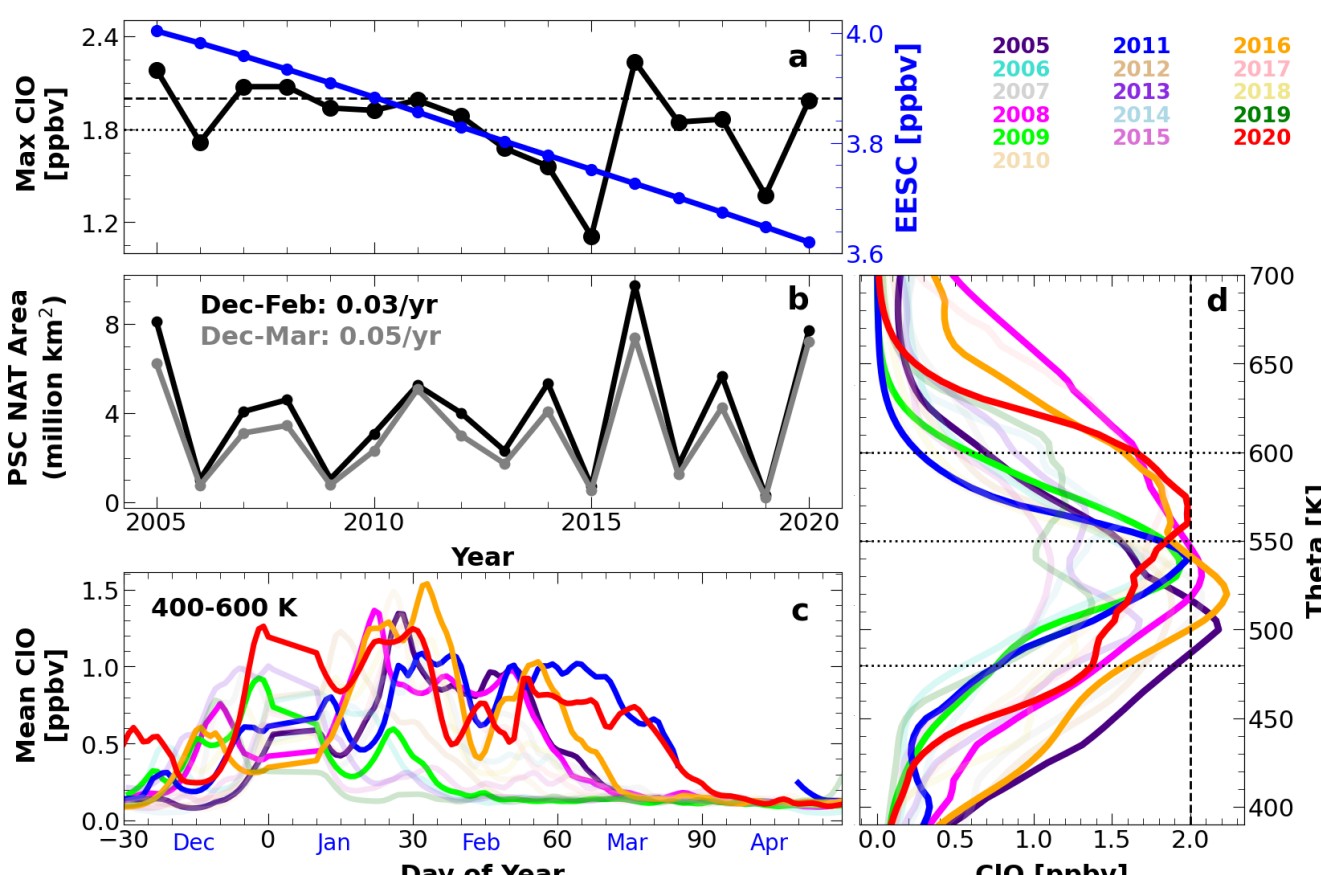

879

**Figure 6: PSC and Chlorine activation the Arctic winters 2005–2020. (a)** The temporal evolution of ClO in the Arctic winters as measured by the Microwave Limb Sounder (MLS) inside the vortex and Effective Equivalent Stratospheric Chlorine (EESC). **(b)** The Area of PSC averaged for the period December-February and December-March (grey) for the winter since 2005. **(c)** The maximum ClO measured inside the vortex in each winter from 2005 to 2020. **(d)** The maximum ClO profiles measured inside the vortex for Arctic winters since 2005. The high chlorine activation with high ClO values are shown in bright colours and others are faded in (a) and (c). Since the chlorine activation timing is different in different winters, the peak ClO observed between December and April/March are shown.

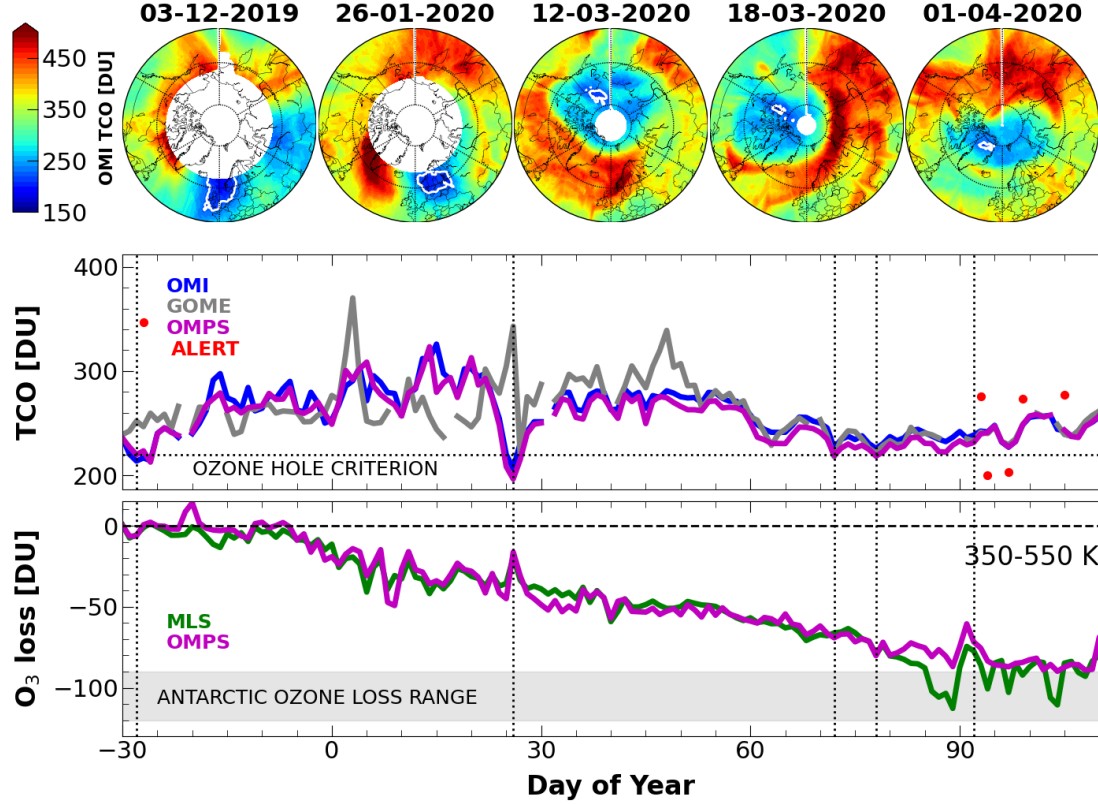

**Figure 7: Arctic ozone in the total column and partial column ozone. Top:** The maps of total column ozone from the OMI satellite measurements in the Arctic for selected ozone hole days for the winter 2020. **Middle panel**: The lowest (5%) TCO measured inside the vortex from three different satellite measurements (OMI, GOME and OMPS). The difference in total column measurements is due to the difference in coverage of the measurements in the Arctic region. The ozone hole criterion of 220 DU is indicated by the dotted line. The total column ozone (TCO) measurements at Alert station are also shown (red solid circles). **Bottom**. The partial column ozone loss computed at the altitude range 350–550 K from the MLS and OMPS measurements. The ozone loss estimated in the Antarctic winters at the same altitude range is shown as the grey-colored area.

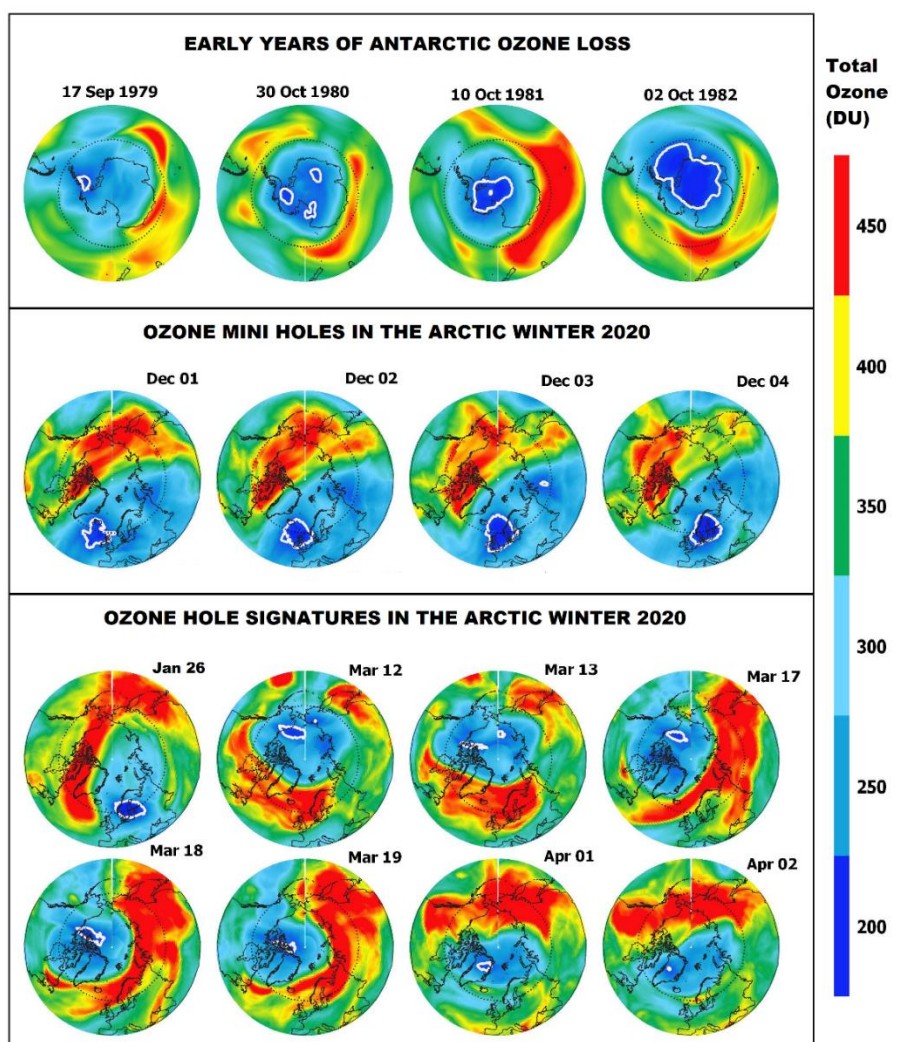

907

**Figure 8:** Maps of total column ozone from MERRA-2 and OMPS satellite measurements for selected days. The Antarctic ozone hole is defined as the area below 220 DU of ozone, as demarcated by the white contour. The top panel shows the early years of the Antarctic ozone hole, the middle panel shows the ozone-mini holes driven by dynamics, and the bottom panel shows the ozone column observed in the Arctic winter 2020.

912

913    JK/final/11082021/

