# Peer review of "Exceptional loss in ozone in the Arctic winter/spring 2020"

_Atmospheric Chemistry and Physics, 2020_

## Community Comment (CC1)

I have two general comments on this manuscript that I think raise important issues that should be addressed before publication:

**Inadequate citation of and discussion of relationships to previously published papers on ozone loss and meteorology in the Arctic 2019/2020 winter:**

There are at least about a dozen peer-reviewed papers already published on the 2019/2020 winter, including one comprehensive overview of the meteorology and its relationships to ozone loss (Lawrence et al 2020) and many that discuss and / or model chemical ozone loss in the Arctic vortex and the record low ozone values.  Only two of these papers (the Manney et al, 2020 and Wohltmann et al, 2020 papers listed) are cited here.  Many, but not all, of these are in the JGR/GRL special issue,
https://agupubs.onlinelibrary.wiley.com/doi/toc/10.1002/(ISSN)1944-8007.ARCTICSPV
In which the first papers were published online in July 2020 and all except two recent ones published in or before November 2020.  All of these contain material that would be useful to cite (though a couple of the dynamical ones possibly only briefly for context) in this paper, and some of them seem critical to cite.  In particular, Lawrence et al (2020) needs to be cited for the discussion of the meteorology leading to the exceptional ozone loss.  The material in Figures 1 and 2 of the current manuscript are, as far as I can tell, completely covered by Lawrence et al (2020), Wohltmann et al (2020), and Dameris et al (2021, ACP, https://doi.org/10.5194/acp-21-617-2021), so if they are to be included in the final paper, the authors need to highlight something that is new in their presentation of the material.  (In that discussion it would also be worth citing DeLand et al (2020) for actual PSC observations.)   All of the following papers include discussion of anomalous column ozone and its implications, and should be cited in addition to Wohltmann et al. (2020):  Rao and Garfinkel (2020), Inness et al. (2020), Bernhard et al. (2020), Dameris et al (2021, ACP), Feng et al (2021), Weber et al (2021).  Several of these papers (as well as Manney et al, 2020 and Wohltmann et al, 2020) include estimates from data and/or modeling of amounts of chemical ozone loss in 2019/2020 in relation to previous years (including especially 2010/2011), and the results in this paper should be discussed in the context of those in these papers, and what is new in this paper clearly highlighted.

**Inadvisably casual use of the term "ozone hole" for the Arctic:**

There are many reasons (first discussed extensively in relation to the 2010/2011 winter, e.g., see Solomon et al, 2014, https://doi.org/10.1073/pnas.1319307111) why one should be very careful and precise about applying the term "ozone hole" to the Arctic.  There is some discussion of this in Wohltmann et al (2020), and I do not want to go through all of the detailed arguments again, so I **strongly** urge the authors of this manuscript to read the reviews of (especially the one by Dr. Wohltmann) and the SC by Grooß & Manney in the discussion of Dameris et al (2021, ACP) for a comprehensive discussion of this point, and use these cautions to consider and revise the presentation of the results in this paper accordingly.

---

## Author Comment (AC1)

We thank the reviewers and Dr. Gloria Manney for their comments and suggestions for improving the paper. Our point-by-point responses to the reviewers' comments are given below in blue text, and the revisions are shown in the version of the manuscript with track changes.
* * *
REPLIES to Referee #2 comments

I struggled with this review because while the paper presents a lot of information, and the authors have done a lot of work, it is difficult to see what is new. The major discoveries announced in the abstract, for example, "Sporadic occurrences of ozone hole values ... record-breaking ozone loss of about 2.0–3.4 ppmv ... unprecedented chlorine activation ... first-ever appearance of loss [near] saturation in the Arctic..." have been made, and published, by Manney et al. (2020) or Woltmann et al. (2020). The title ("on the verge") suggests that the authors think we will see a real ozone hole, or more ozone holes in the future. But they fail to present any dynamical arguments for that. Figure 1a of Woltmann et al. (2020) in fact suggests this, so it would be interesting to see a climate model prediction of the like, or more long-term analysis of trends in dynamical parameters.

Thank you for the comments and suggestions.

The Arctic winter/spring in 2019/2020 is unique in terms of the stratospheric meteorological conditions and its consequences in the very low polar stratosphere ozone, which has also discussed by other studies in JGR/GRL special issue. However, we would like to provide a comprehensive analysis to investigate the polar processing and ozone loss in the Arctic winter/spring 2019/2020 using different observations and reanalysis datasets as well as model simulation, which would help to assess the impact of climate change on winters and aid modelling and forecast of such events in future.

Our study is very important in this context, as we have used six different satellite measurements to describe the chemistry, two different methods to estimate the ozone loss, two different meteorology analyses to explain the dynamics, analysis of the ozone mini-hole situation, multi-satellite measurements of observed very low total column values and observation of saturation of ozone loss. All our results are supported and confirmed by different measurements and methods, though most of the conclusions are consistent with other related studies. We have removed the statement on climate connection.

Instead, the paper waffles back and forth about whether there was an "ozone hole" in 2020, even contradicting itself, e.g., "all the methods, data, and parameters converge to provide an undeniable fact of the first-ever ozone hole" and then "the ozone loss in the Arctic cannot not be called as "an ozone hole"." In any case, this is not an important scientific argument, but a quibble about terminology.

Thanks for the concern. We agree that the "ozone hole" is not an important scientific argument but how to make the definition for the Arctic. However, we have never used any statement that confirms an ozone hole in the Arctic. We have used "?" where the ozone hole is mentioned, even in the title. Also, there is a section on whether there is ozone hole or not. We have clearly stated that the ozone loss in the winter cannot be called an ozone hole. To avoid the confusion, we have removed the ozone hole statements from the conclusion now, **Title, Section 3.8, lines 422-423, 452-454, 476-478**

Figure 3 is very nice, with a lot of important information, but that information is already in Manney et al. (2020). The ozonesondes, and the degree to which loss saturation was approached, are thoroughly presented in Woltmann et al. (2020). Loss saturation was never reached, in fact: Antarctic ozone hole profiles frequently show ozone below the detection limit of the sondes (~1 ppbv), while the lowest observed last spring in the Arctic was 125 ppbv. Sondes can measure ozone levels below 100 ppbv with good accuracy (as they do in the troposphere).

Thanks. Though a similar information about the time series of observed ClO, HNO$_3$, N$_2$O and O$_3$ from MLS is shown in Manney et al. (2020), it is very important to discuss the changes of these species involving the polar processing in the winter related to ozone depletion, both dynamical situation and chemistry. Figure 1 describes the dynamics and Figure 3 shows the chemistry, including ozone loss. We have used two different method and three different data sets together in this study to analyse the chemical ozone loss, which is the novelty of this study. We believe that we need different types of analysis for each winter to assess the chemical and dynamical situation to assess the changes in the region. Including Figure 3 in the current version will also provide important information for readers to understand these processes.

Done. It has to be noted that we have not seen any comparison or literature showing 1 ppbv as ozonesonde detection limit. However, there are studies claiming the detection limit as 10 ppbv, but we agree that the studies have used 1 to 40 ppbv as detection limit in different studies (Solomon et al., 2005; Vömel and Diaz, 2010). Here, we use 10 ppbv for this and 0.2 ppmv as the saturation of ozone loss, which is the loss incurred only by the loss cycles in the lower stratosphere. This is mentioned in **lines 398-399**.

I did, however, find the discussion of the ozone mini-holes in December 2019 and January 2020 quite intriguing, especially the observation that they contained high ClO. Mini-holes are generally regarded as dynamic phenomena, so the suggestion that heterogeneous chemistry is occurring is interesting. It might be interesting to explore this further: are they also becoming more common? Do they affect the overall loss of ozone? See also Stenke and Grewe (2003). I noted that in Figure 7 the mini-holes were the only point where TCO fell below 220 DU. That seems worth exploring.

Thanks. Arctic winter 2019/20 is quite interesting including the miniholes in Decemer 2019 and January 2020.

We covered the mini-hole episode in this paper as we wanted to describe the polar processing in the winter thoroughly. In addition, this mini-hole was particularly important as it was slightly modified by the chemistry we have done the trajectory and ClO analysis to complete analyses on this year's mini-ozone hole episode. Anything beyond this demands substantial work on modelling, which is beyond the scope of this study.

Yes, the early winter ozone hole values were related to ozone mini holes, and we have clearly mentioned them. However, as suggested, we have given a brief description of mini-holes that occurred in the past, in **lines 236-248.**

I also appreciated the long-term comparison with previous years in Figure 6. Perhaps this could expanded, along with an analysis of the long-term changes in vortex temperature, V_PSC, wave disturbances/stability…

Thanks for the suggestion. It is included now. Although the long-term analysis is presented in **Figure 1 and Figure S2**, we have made another long-term analysis with all dynamical parameters and presented in **Figure S1b** as suggested. The discussion is given in **lines 177-181.**

Minor points:

Line 23: "provided the stratospheric chlorine levels still stay high there." I don't think there is much uncertainty about future Cl levels.

Done. We have corrected this in **lines 21-22.**

Line 34: "because"? Perhaps "possibly because". This is far from certain, or we wouldn't still be producing ozone assessments. In fact a lot of data show the opposite (decline since 1997).

Done. We meant that it is very difficult to estimate or find a trend in the Arctic data as it is very difficult to differentiate the dynamical contribution from the data. We have changed the sentence now. Please find it **in line 34**.

Lines 58-59: "Here, we show that the Arctic winter in 2020 ... met the condition for an ozone hole for the first time". What condition? This disagrees with most other assessments (e.g. Woltmann et al., 2020; Manney et al., 2020; Wilka et al., 2021).

Done. We have rephrased this. We have removed the term ozone hole" and changed it to "very low ozone" values below 220 DU were there for more than 23 days, and the ozone loss was also record-breaking in the lower stratospheric altitudes below 500 K. This is the situation. All three references mentioned in the comments agreed to this situation, but they were not using the word "ozone hole", although Wilka et al. used the word, as we did in this work.

We have reprahsed the sentences that use "ozone hole" now, including title, Please find the revised **Title, Section 3.8, lines 422-423, 452-454, 476-478**

Lines 60-65: Should indicate where the data were obtained. Uncertainties are quoted but no citation is given.

Done. Data details are given in *Data availability* statement. Citation is mentioned **in line 81.**

Line 80: "The missing values in satellite measurements were filled with linear interpolation (poison_grid_fill)." What is "poison_grid_fill"? How does it work? What are the criteria used for filling?

Done. This is a function with python for the linear interpolation. We have deleted it because it is a method for the data processing.

Line 120: A lot of this paragraph is confusing, but this line especially. T_NAT is 195, not 200 K.

Done. Please note that we were not talking about T_NAT, but the temperatures below 200 K in the winters, which were uncommon during the period. However, note that temperatures below 200 K also include temperatures below 195 K, which is why PSC temperatures are also mentioned. We have rephrased this in **lines 132-134** to make it clearer.

Line 133-134: "This is the largest ice PSC ever observed in terms of its area, volume and number of days of appearance (i.e. frequency) in the Arctic and the area is twice that of the winter 2011." So what? This information is never used for anything.

Done. It was showing the extreme meteorological situation in the winter and was also the reason for the large loss in the lower stratosphere as compared to 2011. This is mentioned in **lines 338-339**.

Lines 151-156: This is interesting, potentially, but vague and hand-waving. It could be really valuable to have an analysis that looks at the evolution and variation of the Arctic vortex over the last 20+ years.

Done. This is shown **in Figure S1B** for the past 41 years and described in **lines 177-181**.

Lines 210-212: This analysis might make an interesting paper, if expanded.

Thanks. We have analysed the ozone mini-hole in the winter, its temporal evolution, air mass transport, and analyses with ozone measurements by satellite and ozonesondes. Anything beyond this is modelling of mini-holes, which is beyond the scope of this paper, and warrants a separate study and we would do that. However, we do respect the suggestion, and we have presented a brief description of the mini-holes, in **lines 236-248.**

Lines 360-364: This interpretation is incorrect. The sondes do indeed have an uncertainty of about 10%, but that means that the minima of 0.125 (or 0.200) ppmv would have error bars of ±0.012 (or ±0.020). That is not consistent with zero, or even 0.1 ppmv.

The 5% accuracy is noted from the ozonesonde JOSIE intercomparison project and Smit et al. (2017). We could not find any document stating this 10% uncertainty. We have, however, stated this as 5-10% in **line 406.**

Reference: Stenke, A., and V. Grewe (2003), Impact of ozone mini-holes on the heterogeneous destruction of stratospheric ozone, *Chemosphere, 50*, 177-190, https://doi.org/10.1016/S0045-6535(02)00599-4

Thank you. We have cited this paper, **lines 229, 247.**

---

## Author Comment (AC2)

We thank two reviewers and Dr. Gloria Manney for their comments and suggestions for improving the paper. Our point-by-point responses to the reviewers' comments are given below in blue text, and the revisions are shown in the version of the manuscript with track changes.

REPLIES TO Referee #1 Comments

In their manuscript, Kuttippurath et al. investigate Arctic stratospheric ozone loss during the exceptional winter 2019/2020 from a range of satellite and ground-based observations. Their analysis is thorough and the results are sound. It is, however, less clear to me, what the main message of this paper is. Previous studies, correctly cited in this manuscript, have already come to similar results. So it would be good, if Kuttippurath et al. could spell out a bit clearer what this study adds, that is not already known from these previous studies.

I have one concern with the claims made here that the exceptional ozone loss in 2019/2020 is a sign of climate change. As far as I am aware of the current literature, most climate models do not show any increase in Arctic ozone loss due to climate change. Can the authors rule out that 2020 was not just an extreme winter within the current range of variability? And related to that point: the Arctic did experience in March 2020 ozone hole conditions, as this study demonstrates. Is there evidence of an "exposure of nearly 650 million people and ecosystem to unhealthy ultra-violet radiation levels" (quoting from the first sentence of the abstract)? Or do the authors suggest that future Arctic winters could show even larger ozone depletion? And if so, on which basis? The authors should try to make these points clearer.

Thanks for the comments

We would like to emphasize that we wanted to describe the polar processing and ozone loss in the winter 2020, as agreed by the reviewer that "Their analysis is thorough and the results are sound". We agree that there are studies on this winter, as mentioned. However, it is important that we need multiple studies using different data and model simulations on different as aspects the winter, which would help assess the winter to assess the impact of climate on winters and aid modelling and forecast of such winters in future (please also see our replies to referee #2). We have removed the statement on climate connection now.

The paper is overall generally well written, but could be made clearer at several points. See my specific comments below. I recommend the manuscript for publication in Atmos. Chem. Phys. if the authors can address my general and specific comments.

Thank you.

*Specific comments:*

P1, l13: "Severe vortex-wide ozone loss in the Arctic would expose nearly 650 million people and ecosystem to unhealthy ultra-violet radiation levels." The number of 650 million people does not appear in the body of the manuscript and is not backed-up by any citation. So I suggest removing this explicit statement here from the abstract.

Done. We have removed the sentence in the abstract as suggested.

p1, l22: "the very colder Arctic winters in the near future will experience even more ozone loss": do you mean Arctic winters in the near future will become colder? On what basis is this claim made? Or do you mean the coldest Arctic winters in the near future within the current range of variability? But why should they experience very likely even larger losses?

Done. We meant that it is very likely that the colder winters might get more colder. We have removed the climate connection statement and "cold winters colder with more ozone loss" speculation in lines 21-22.

p1, l22: language: "Our study suggests that the very colder Arctic winters in near future would also very likely to experience even more ozone loss and encounter ozone hole situations, provided the stratospheric chlorine levels still stay high there." -> "Our study suggests that colder Arctic winters in the near future would likely experience even more ozone loss and encounter ozone hole situations, as long as stratospheric chlorine levels remain high."

Thanks. Done. Rephrased as suggested, lines 21-22.

p1, l30: why did the Antarctic ozone loss peak in the late 1980s when polar stratospheric chlorine loading peaked around the early 2000s

There were already sufficient chlorine in the Antarctic stratosphere to reach the loss saturation. Therefore, even if there is more chlorine in the stratosphere that wouldn't affect the ozone loss saturation anymore, but the ozone loss will be modified by the inter-annual changes in meteorology of the winters. This is mentioned in line 30.

p2, l42: "e.g., > 1.5ppmv of loss" seems arbitrary. Please motivate this value

Done. This is the average ozone loss value taken from different literature that have multi-year analyses. Please find the revised statement with references in lines 42-43.

p2, l43: 25-30% in which metric? The statement on 1.5ppmv above clearly refers to loss at a certain altitude. On a given altitude, previous Arctic winters (such as 1999/2000 or 2010/11) experienced losses far greater than 25-30% (e.g., Sinnhuber et al., 2000; Sinnhuber et al., 2011). Please be specific which metric this refers to: column loss with the vortex, column loss poleward of a certain latitude, local loss,

Done. These are mostly from satellite (Livesey et al., Kuttippurath et al., Manney et al.), sondes (Rex et al., ) and ozone column from the ground (e.g. SAOZ). The source and respective reference with rephrased statements can be found in lines 43-48

p2, l46: "short-lived" is what sense?

 Done. The Arctic vortex is mostly in tact from December through mid-February or early March. This is what we meant by short-lived. This is short as compared to December through end of April vortex this year. This is mentioned in lines 50-51.

P2, l50: "ozone loss is found to be proportional to the timing of the major warmings": I think I know what you mean, but this statement is not very clear

Done. We mention that the vortex longevity is a factor for prolonged ozone loss, although 1997 is an exception. This is rephrased in lines 57-58.

P2, l53: "The occurrence of extreme events is a signature of climate change and so are the extreme cold winters with large loss in ozone (e.g. IPCC, 2007)" Sorry, it may be true that under a changing climate the occurrence of extreme cold winters may increase, but it is not at all clear if there is a trend towards more extreme events in Arctic stratospheric temperatures and whether or not it is related to climate change! This statement is not backed-up by IPCC, 2007.

Done. As mentioned earlier, we have removed the climate connection statement, as it was confusing. The IPCC report was quoted for the occurrence of extreme whether events as polar meteorology is a key element for sustained ozone loss. Please find the revised statement in lines 21-22, 64-65.

p2,l62: would be good to have references for the data sets

Done. Please find them in lines 107, 112, 114 and 115

p2,l64: latitude and longitudes swapped for Alert

Sorry for this. Corrected in lines 80-81.

p2,l65: Do the 5-10% refer only to the sondes, or also to the satellite profile data?

This is mentioned for the satellite measurements here. This is mentioned in line 81.

p3, l70: GOME -> GOME-2 ?

Done. Please find it in line 86.

p3, l74: "and other trace gas profiles": which?

Done. Other trace gases are ClO, $N_2O$, and $HNO_3$. This is mentioned in line 99.

p3, l75: does OMPS provide temperature profiles?

This is mentioned here for MLS and for OMPS. We have used the ozone profiles and column value provided, mentioned in line 92.

p3, l80: what is poisson_grid_fill ? Reference?

Done. We are Sorry. This is a normal linear interpolation procedure done with Python software. We have removed it now since it is a function in python for the data processing.

p3, l84: if the precision varies so strongly, maybe better to give percentage uncertainty?

We have mentioned the uncertainty in absolute values because there are biases as well, as suggested by the validation papers. Thank you.

p3, l87: not clear to me how well justified this extrapolation is. Does this extrapolation takes into account the tropospheric N2O VMR?

Done. Yes, it accounts for tropospheric values too, mentioned in line 104.

p4, l114: "longest winters"? You mean latest vortex break-up? Or coldest winters?

Done. We meant the winters with long-lasting vortex up to April, mentioned in line 130.

p4, l129: "PSC area" -> "potential PSC area". Please make clear, that this is not area of observed PSCs but area of temperatures cold enough for formation of PSCs.

Done. This is rephrased in line 149.

p5 , l133: "in 40 years": where there colder temperatures before, or are these the coldest ever observed?

Done. Yes, as per the MERRA-2 data as shown in Figure 1.

p5, l133: "the largest ice PSC ": This is likely not a single cloud, but an area of temperatures cold enough for the formation of ice PSCs

Done. Yes, corrected in line 153.

p5, l143-150: This general discussion of the relation between wave activity and vortex strength can be moved to the introduction.

Done. This is now in Introduction in lines 51-58.

p5, l159: "occupied the entire polar region": how do you define polar region? North of 60N? Or entire vortex?

Entire vortex, mentioned in line 173.

p6, l165-168: can be removed, redundant

Done. Removed.

p6, l188: didn't Rex define APSC and VPSC as the temporal integral of PSC area and volume, respectively?

Done. This is mentioned in line 208.

p7, l196: From Fig. 3: I don't see a gradual descent of loss from the middle stratosphere to the lower stratosphere: I see some (small) loss above 600K in December and much larger losses in the lower stratosphere (below 600K) beginning in December as well and intensifying during January. Or does this statement refer to earlier winters?

In fact, there will be ozone loss at higher altitude in December (below 0.5 ppm) and then progresses to lower stratosphere as the winter/spring advances. This is common to all winters, but there will be differences in values of ozone loss. This is what we mentioned in lines 217-218

p7, l200: chlorine activation does not require sunlight, but high levels of ClO do

Yes, this is rephrased for clarity in line 220.

p7, l216: the high levels of ClO in air masses with low PV are very surprising. The high ClO suggests that the reductions are not only "dynamically driven"? Would be great to have a bit more discussion at this point.

Done. We have added a small description of the process in lines 230-234.

p7, l220: citations seem out of place

Done, corrected in **line 256.**

p9, l264: "chlorine activation and ozone loss is limited to the winters with very low temperatures in December–February" this statement is somewhat incorrect

Done. Yes, ozone loss can always be there, but the loss is significant or not is the point. This is rephrased in **line 301.**

p9, l268: "ozone loss in the winter 2011 was about 1.0 ppmv (or 30–40 DU),": I don't understand what these numbers refer to. E.g., Sinnhuber et al., 2011, derived maximum ozone loss in Arctic winter 2010/11 of more than 2 ppm at 19 km and more than 120 DU column loss. Is this what is meant in the next sentence: "which is higher than that of other Arctic winters (about 2.1–2.3 ppmv or 100–100 DU)"? (100-100DU is a typo anyway, I guess.) Is the first sentence then referring to loss before February only?

Done. These measurements are mostly from satellite (Livesey et al., Kuttippurath et al., Manney et al.), sondes (Rex et al., ) and ozone column from the ground (e.g. SAOZ). The source and respective reference with rephrased statements can be found in **lines 42-48, 302-307**

p9, l77: "undoubtedly" is a strong word. I suggest to remove.

 This is removed now.

Fig. 5: I couldn't find for which period in the given years the ClO profiles are shown. Are these maximum values or temporal averages? Any idea why ClO is so much higher above 550 K in the Antarctic in 2019 compared to 2015 – keeping in mind that 2019 was a rather warm and disturbed Antarctic spring?

Done. The shown profiles are selected such that they represent the maximum observed ClO in each winter. Then we have averaged the profiles three days (max day +/-1 day) to avoid any error in single day measurements. This is mentioned in **lines 828-829**.

The ClO profiles have a broader peak in the Antarctic; it is not only for 2019, but for all Antarctic winters, because of the meteorology and strong Chlorine activation there. This can be seen in Kuttippurath et al. (2015), who have done the analyses for 10 Antarctic winters here. More detailed analysis is needed for this Antarctic winter. We will do that in a separate paper. Thank you.

Section 3.5, Fig. 6: When discussing the maximum ClO amounts in the past winters, it would be interesting to put this into context of the EESC (or similar metric): By how much has total chlorine (or EESC, …) decreased between 2005 and 2020?

Done. Its 246 ppt/year. This is mentioned in **lines 343-346**.

 p11, l346: how exactly is saturation ("complete loss of ozone") defined here? In reality ozone is of course never completely gone. Okay, I see further down at l359 that you define this as below 200ppbv with a reference to Smit et al. (2007). I believe it would be good to include a brief justification here, why this is a useful definition for loss saturation.

Done. The definition is depending on the detection limit. This detection limit is 10-20 ppbv. An explanation is given in **lines 398-399**. The reference is also given there.

p11, l347: Again, I don't understand the meaning of "ozone loss normally happens only up to 25–30% in the Arctic winters". Local loss in previous cold Arctic winters was clearly larger than 30%.

Done. These are taken from Gautail et al., and Pommereau et al. Yes, these are from the ground-based spectrometer measurements. This is the only long-term ozone loss estimate available in the Arctic, which is why these are mentioned. This is rephrased in lines 434-435

Fig. 5: Please indicate the dates for the sonde profiles.

Done. Please find the dates in figure caption (8 April and 10 April 2020), line 833.

p12, l362: "the loss saturation suggests that the Arctic has entered an exigent climate change scenario": again, it is not self-evident, why this is a sign of climate change and not just an extreme winter within the range of variability. Same comment applies to l403.

Done. We have removed the sentence.

p12, l380: Just for curiosity: why are GOME measurements more restricted in latitude than OMI or OMPS? I thought all three use similar wavelengths ranges?

It depends on the orbit and elevation of satellite.

p12, l389: contradiction: in the previous sentence it is stated that a column loss of about 90-120 DU occurs in extreme winters such as 2005 and 2011 and in the next, the largest observed loss was 100 DU in 2011. Sorry, but these small contradictions are very confusing and make for a tiresome reading.

Done. Sorry for that. This is rephrased and corrected now Griffin et al., and Livesey et al. have complied the loss estimates for different winters. Average values from those studies are shown here. . Please find it in lines 436-437.

P14, l437: "Extreme weather events are harbingers of climate change": See comments above on extremes and climate change.

The sentence is removed.

Technical corrections

p2, l61: ->"We have used two satellite ozone profile datasets."

Done. Please find it in line 77.

p2, l77: ER5 -> ERA5

Done. Please find it in line 92.

p2, l78/79: active and passive voice changes

Done. Please find it in lines 94-95.

p6, l186: ozone AND N2O

Done. Please find it in line 205.

p7, l227: present IN all

Done. Please find it in **line 263.**

p9, l286: Wohltmann

Done. Please find it in **line 324**.

p12, l369: by Nash et al.

Done. Please find it in **line 416**.

p13, l404: not present continuously

Done. Please find it in **line 453**.

---

## Author Comment (AC3)

We thank the reviewers and Dr. Gloria Manney for their comments and suggestions for improving the paper. Our point-by-point responses to the reviewers' comments are given below in blue text, and the revisions are shown in the version of the manuscript with track changes.
* * *
REPLIES TO Gloria Manney Comments, 31 Mar 2021
* * *
I have two general comments on this manuscript that I think raise important issues that should be addressed before publication: Inadequate citation of and discussion of relationships to previously published papers on ozone loss and meteorology in the Arctic 2019/2020 winter:There are at least about a dozen peer-reviewed papers already published on the 2019/2020 winter, including one comprehensive overview of the meteorology and its relationships to ozone loss (Lawrence et al 2020) and many that discuss and/ or model chemical ozone loss in the Arctic vortex and the record low ozone values. Only two of these papers (the Manney et al,2020 and Wohltmann et al, 2020 papers listed) are cited here. Many, but not all, of these are in the JGR/GRL special issue,https://agupubs.onlinelibrary.wiley.com/doi/toc/10.1002/(ISSN)1944-8007.ARCTICSPV

In which the first papers were published online in July 2020 and all except two recent ones published in or before November 2020. All of these contain material that would be useful to cite(though a couple of the dynamical ones possibly only briefly for context) in this paper, and some of them seem critical to cite. In particular, Lawrence et al (2020) needs to be cited for the discussion of the meteorology leading to the exceptional ozone loss. The material in Figures 1 and 2 of the current manuscript are, as far as I can tell, completely covered by Lawrence et al(2020), Wohltmann et al (2020), and Dameris et al(2021, ACP, https://doi.org/10.5194/acp-21-617-2021), so if they are to be included in the final paper, the authors need to highlight something that is new in their presentation of the material. (In that discussion it would also be worth citing DeLand et al. (2020) for actual PSC observations.) All of the following papers include discussion of anomalous column ozone and its implications, and should be cited in addition to Wohltmann et al. (2020):Rao and Garfinkel (2020), Inness et al.(2020), Bernhard et al. (2020), Dameris et al (2021,ACP), Feng et al (2021), Weber et al(2021).

Several of these papers (as well as Manney et al, 2020 and Wohltmann et al, 2020) include estimates from data and/or modeling of amounts of chemical ozone loss in 2019/2020 in relation to previous years (including especially 2010/2011),and the results in this paper should be discussed in the context of those in these papers, and what is new in this paper clearly highlighted.

Thanks for the comments. We are really sorry for having not cited enough papers published in the JGR/GRL special issue because our original manuscript was first submitted last June and was delayed after we have tried other journals. Therefore, we cited then available two published papers, Manney et al., and Wohltmann et al. These papers came after we submitted the paper to other journals and some were still not published. There it was not deliberate that we left out some.

We have included the latest studies on this winter and cited them wherever is appropriate. We have not cited any discussion papers, but only peer-reviewed. Please find the cited ones in lines 70, 71, 154, 168, 174, 175, 187, 270, 315-316, 402-403, and 467. The citation details can be found in the reference section.

Inadvisably casual use of the term "ozone hole" for the Arctic:

There are many reasons (first discussed extensively in relation to the 2010/2011 winter, e.g., see Solomon et al, 2014, https://doi.org/10.1073/pnas.1319307111) why one should be very careful and precise about applying the term "ozone hole" to the Arctic. There is some discussion of this in Wohltmann et al (2020), and I do not want to go through all of the detailed arguments again, so I strongly urge the authors of this manuscript to read the reviews of(especially the one by Dr. Wohltmann) and the SC by Grooß & Manney in the discussion of Dameris et al (2021,

ACP) for a comprehensive discussion of this point, and use these cautions to consider and revise the presentation of the results in this paper accordingly.

Thanks for the suggestion.  We agree that it will be confusion to term Arctic ozone hole since this is not happening every year in the Arctic. Therefore, we have revised the discussion accordingly. Please find it in Title, Section 3.8 and elsewhere in the text and replies to other two referee comments. Thank you.

Thank you for your critical comments that helped to improve the content of this article.

---

## Author Comment (AC5)

**Supporting information for**

**Exceptional loss in ozone in the Arctic winter/spring 2020**

Jayanarayanan Kuttippurath[1]*, Wuhu Feng[2,3], Rolf Müller[4], Pankaj Kumar[1], Sarath Raj[1],

Gopalakrishna Pillai Gopikrishnan[1], Raina Roy[5]

[1]CORAL, Indian Institute of Technology Kharagpur, Kharagpur—721302, India.

[2] National Centre for Atmospheric Science, University of Leeds, Leeds, LS2 9PH, UK

[3] School of Earth and Environment, University of Leeds, Leeds, LS2 9JT, UK

[4]Forschungszentrum Jülich GmbH (IEK-7), 52425 Jülich, Germany

[5]Department of Physical Oceanography, Cochin University of Science and Technology, Kochi, India
* * *
This document provides Supporting Information for the main ACP paper. This information consists of four supplementary figures, which provide further information about the stratospheric meteorological conditions since 1979 to 2020 or present results for additional ozone and $N_2O$ correlation, ozone loss and observed ozone evolution from ozonesonde measurements.

**Figure S1** shows the time series of area of PSC (APSC) and volume of PSC (VPSC) at 460 K (~50 hPa) in the Arctic winters from 1979 to 2020, estimated from the MERRA-2 reanalysis dataset. APSC and VPSC are calculated using the definition from Rex et al. [2005]. Shaded range is their standard deviation from the mean.

**Figure S1b** is the temporal evolution of Dec-Feb (black line) and Dec-March (grey line) averaged temperature, zonal winds, vortex area, and heat flux in the Arctic winters from 1979 to 2020, as estimated using the MERRA-2 data. The linear trend values for these variables are calculated and given in the corresponding panels.

**Figure S2** shows the time evolution correlation between observed ozone and $N_2O$ inside the polar vortex for two cold Arctic winter/spring 2010/11 and 2019/20.

**Figure S3** is the derived monthly averaged ozone loss with its standard deviation (horizontal bars) from 350-700 K inside the polar vortex based on the tracer descent method.

**Figure S4** shows the temporal evolution of observed minimum ozone above 400 K by the ozonesonde at Eureka and Alert stations.

**Table 1** lists total days when the observed total column ozone value is less than 220 DU anywhere inside the polar vortex from OMPS satellite measurement and MERR-2 reanalysis dataset.

[Figure]

**Figure S1:** The temporal evolution of area of PSC and volume of PSC in the Arctic winters from 1979 to 2020, as estimated using the MERRA-2 data. The shaded area is the standard deviation from the mean.

[Figure]

**Figure S1b:** The temporal evolution of temperature, zonal winds, vortex area, and heat flux in the Arctic winters from 1979 to 2020, as estimated using the MERRA-2 data.

[Figure]

**Figure S2:** The time evolution correlation between ozone and $N_2O$ in the Arctic winter 2020. The measurements selected inside the vortex.

[Figure]

**Figure S3:** The monthly averaged ozone loss and the standard deviation (horizontal bars) computed using the tracer descent method.

[Figure]

**Figure S4:** The temporal evolution stratospheric ozone as observed by the ozonesonde at Eureka and Alert stations.

Table 1: Total days when the observed total column ozone value is less than 220 DU anywhere inside the polar vortex

*OMPS (24 Days)*

- Dec 01 − 05 (5 Days)
- Jan 01 − 02 (2 Days)
- Jan 23, 25 − 30 (7 Days)
- Mar 05, 12 − 19, 28 (10 Days)

*MERRA-2 (19 Days)*

- Dec 01 − 05 (5 Days)
- Jan 25 − 26 (2 Days)
- Mar 05, 12, 17 − 22 (8 Days)
- Apr 06 − 07 (2 Days)

---

## Author Response (AR2)

In their revised manuscript, Kuttippurath et al. have addressed the earlier comments. Most importantly, it wasn't really clear what this study adds, that was not known from other published studies on the Arctic winter 2019/20. Nevertheless, the present study is scientifically sound, appropriately acknowledges earlier studies and provides a comprehensive overview. So all in all I recommend publication after consideration of some additional, most minor, comments:
We thank the reviewers for the positive comments and suggestions for improving the paper. Our point-by-point responses to the reviewer comments are detailed below in blue text, and the changes are shown in the version of the manuscript with track changes.

Specific comments:

l.64: "The occurrence of extreme events is a feature of climate change": I'm having problems with this statement as this is too general here: There are always extreme events, even without climate change. I suggest to drop the new sentences "The occurrence of extreme events is a feature of climate change (e.g. IPCC, 2007). Therefore, the extremely cold winters with large loss in ozone could also be a harbinger of climate change." And continue as before with "Previous studies have postulated …", which is much more to the point.
Done, the statement is removed as suggested by the referee.

l.71: "However, in this study, we use different data sets, different loss estimation methods, and several assessment parameters together to study the polar processing and ozone loss in the Arctic winter 2020, and such an analysis is never done for this winter. This is particularly important as the winter was very cold with the largest ozone loss in the observational record and experienced the total column ozone (TCO) values below 220 DU for several days in the vortex, for the first time."

I suggest to avoid statements like "never done" and "for the first time" and just state (matter of fact): "In this study, we use different data sets and several parameters together, to investigate the large ozone loss in the Arctic winter 2020, and the conditions that led to the record low total column ozone (TCO) values below 220 DU in this winter." (Or similar)
Done. We have removed the statement and revised as suggested by the referee in lines 71-73

l.252: The finding of high amounts of ClO in air masses associated with the mini holes are very surprising, and I believe it was already suggested in the previous review round that it may be worth to further investigate that. Unfortunately, this is not further addressed here.
Done. We are very sorry that we cannot discuss this further in this paper as it involves more detailed analysis. This paper is already a very long (13000+ words and 8 figures). As suggested, we would do a separate study in this regard. We hope that the referee and the editor would find it as an appropriate decision. Thank you.

On the other hand, similar studies were also done in the past on ozone mini-holes. For instance, Weber et al. (2002) state the temperature can be below PSC threshold and thus, high ClO can be there. Feng (2006) also demonstrate large areas of PSCs and high ClO in the mini-hole region. These are mentioned in lines 258-260.

l.451: "The appearance of a threshold in TCO below 220 DU for several weeks demonstrates that Arctic winters may enter a new era of ozone depletion events, and signal significant changes in the climate of the region": I don't agree with this statement. It is fine to say that the Arctic may have entered a new level of ozone depletion, but whether or not this signals significant climate change or

may be seen as a new era remains to be shown. This is not supported here in this study or by the given references.

Done. We have deleted " … and signal significant changes in the climate of the region" in lines 448.

Minor comments

l.30 "Montreal protocol" -> "the Montreal protocol"

Done. Please find it line 30

l.40: "high temperatures" -> "higher temperatures"

Done. Please find it line 40

l.45: "the ground-based" -> "ground-based"

Done. Please find it line 45

l.45-48: "ground -based measurements show about 15–20% of loss in most Antarctic winters", I'm confused at this point: is this a typo and should read "in most Arctic winters"?

Done. Thank you for correcting this. Please find the "Arctic" in line 46

l.81: I suggest to move the statement on the uncertainty of the ozone sonde measurements to the end of the next paragraph, after the statement on the satellite uncertainty. Maybe also moving the list of species from MLS up to l.77/78.

Done. The specie added to line 78 and the ozonesonde uncertainty added to the next paragraph end, in line 108

l.163-168: "The diagnosis with heat flux and the eddy heat flux associated with waves demonstrates that the momentum transported from the troposphere to stratosphere was very weak in 2020 (in the range of -20 to 30 Km s -1), and the heat flux values are zero or negative (e.g. -10 Km s -1 in February) during most part of the winter. These results are also in agreement with the eddy heat flux computed for the waves, as they also show smaller wave momentum to the stratosphere. In short, the eddy heat flux and wave heat flux show smaller values in January-April; indicating the reason for the less disturbed long-lasting vortex in 2020."

I find these lines confusing. As I understand both panels in Fig.1 show eddy heat flux, one calculated for all waves, the other for waves 1-3, right? Maybe you could make this clearer.

Done. As suggested, we have specifically mentioned the total or net heat flux and eddy heat flux associated with planetary waves 1, 2, and 3 in lines 163-168

l.194: "We use the profile descent method using the trace of air motions N2O and is a widely used method for ozone loss estimation": This sentence is not very clear.

Done. We have revised it as "We estimate the descent rate from the tracer $N_2O$ inside the polar vortex, then assume the averaged profile descent rate is identical to the dynamical ozone, so that the chemical ozone loss can be derived (e.g., Griffin et al., 2019). This is a widely used method for chemical ozone loss estimation. Please find it in lines 195-197

l.221: "and there is no sunlight in the Arctic vortex in early winter": this statement contradicts what is said a few lines further down. Maybe write "…and one would not expect significant amounts of sunlight in the early winter Arctic vortex…"

Done. Thank you and we have revised as suggested in line 221

l.252: "TCO transported": better say "ozone transported" to link to the discussion on profile changes in the next lines.

Done, please find it in line 253.

l.315: "A very similar conclusion is also presented in the study of Grooß and Müller (2021)." ->
"…higher than that of any previous Arctic winter (Grooß and Müller, 2021)."
Done. Please find it in line 315.

l.338: "record-breaking ice PSCs" -> "record-breaking extent of ice PSCs" (or whatever quantity you are referring to here)
Done. Please find it lines 337-338

l.343: "We also looked at the changes in EESC during the period (2005-2020) and there has been continuous decline in EESC during the period (Fig. 6, top panel). The rate of change of EESC during the period is about 246.16 ppt per year (e.g. WMO, 2018); suggesting a consistent reduction in stratospheric halogen loading in 2020 (e.g. Grooß and Müller, 2021)." This is a bit wordy and can be expressed more compact as "During the period 2005-2020 there has been a continuous decline in EESC by about 246.16ppt per year (e.g. WMO, 2018)." However, how can WMO (2018) be cited for changes up to 2020? I don't understand why this suggests a consistent reduction in halogen loading: this is the change in stratospheric halogen loading. Or do I miss something here?

Done. We have revised the text as suggested by the referee now in lines 343-345. "We also looked at the changes in EESC during the period (2005-2020) and there has been a continuous decline in EESC during the period (Fig. 6, top panel). The predicted rate of change of EESC during the period is about 246.16 ppt per year (e.g. WMO, 2018); suggesting a reduction in stratospheric halogen loading in 2020 compared to the peak loading by about 10% (e.g. Grooß and Müller, 2021)."

l.436: "column ozone loss ever measured": I'm having problems with the statement, that the column ozone loss was measured, as we cannot directly measure column ozone LOSS. We can only measure ozone columns or profiles and derive chemical loss using certain assumptions (as described in this paper).
Done. Agree, it should be estimated or calculated. Rephased as "deduced hitherto" in line 434